# Selective Ensembles for Consistent Predictions

**Emily Black, Klas Leino , Matt Fredrikson**
{emilybla, kleino, mfredrik} @cs.cmu.edu
Carnegie Mellon University

## Abstract

Recent work has shown that models trained to the same objective, and which achieve similar measures of accuracy on consistent test data, may nonetheless behave very differently on individual predictions. This inconsistency is undesirable in high-stakes contexts, such as medical diagnosis and finance. We show that this inconsistent behavior extends beyond predictions to feature attributions, which may likewise have negative implications for the intelligibility of a model, and one's ability to find recourse for subjects. We then introduce *selective ensembles* to mitigate such inconsistencies by applying hypothesis testing to the predictions of a set of models trained using randomly-selected starting conditions; importantly, selective ensembles can abstain in cases where a consistent outcome cannot be achieved up to a specified confidence level. We prove that that prediction disagreement between selective ensembles is bounded, and empirically demonstrate that selective ensembles achieve consistent predictions and feature attributions while maintaining low abstention rates. On several benchmark datasets, selective ensembles reach zero inconsistently predicted points, with abstention rates as low 1.5%.

## 1 Introduction

Recent work has drawn attention to the fact that models that appear similar from aggregate quality measures, such as accuracy, often have markedly different behavior at the level of individual predictions (Black and Fredrikson, 2021; Marx et al., 2019). Further, in deep models, this inconsistency can arise even between closely-related models, such as those arising from different initializations, or from leave-one-out differences in the training data (Black and Fredrikson, 2021; D'Amour et al., 2020). This behavior is undesirable in many high-stakes contexts, such as medical applications and credit-approving scenarios, as it may cast doubt on the justifiability of the model's outcome and pose difficulties for reproducibility and comparison.

We begin by demonstrating that not only are the predictions of related deep models often dissimilar, but their *feature attributions* (Simonyan et al., 2014; Sundararajan et al., 2017; Leino et al., 2018) are as well (Section 3). In particular, we show that there is little connection between a model's gradients, which are the basis for many deep attribution methods, and the labels that it predicts—models with identical predictions can have arbitrarily different gradients almost everywhere (Theorem 3.1). In practice, we show that this result occurs often on common datasets across closely-related models, leading to significant variation in attributions. This may be undesirable, as feature attributions are commonly used to provide explanations (Simonyan et al., 2014; Sundararajan et al., 2017; Leino et al., 2018), debug model behavior (Adebayo et al., 2020), and diagnose problems related to privacy and fairness (Leino and Fredrikson, 2020; Datta et al., 2016). Beyond these pragmatic concerns, this suggests that the salient factors behind these models' predictions on many points may have little in common, even when models appear to do comparably well on test data.

To address inconsistency in both prediction and attribution, we then turn to ensembling, a well-known approach for reducing predictive variance (Meir et al., 1995; Naftaly et al., 1997; Lincoln and Skrzypek, 1990; Fumera et al., 2005; Hansen and Salamon, 1990; Krogh and Vedelsby, 1995). We introduce *selective ensembles*, which leverage a recent result on multinomial rank verification (Hung et al., 2019)—which has also been used recently for making certifiably-robust predictions (Cohen et al., 2019)—to efficiently mitigate the problem of inconsistency with a probabilistic guarantee. Given a point to classify, a selective ensemble returns the mode of the class labels predicted on that point, where the mode is sampled over models that vary according to a specified source of randomness in the training process. Importantly, if the mode cannot be inferred with sufficient confidence, then the selective ensemble *abstains* from prediction. This allows us to bound the probability that these ensembles do not return the true mode prediction (Theorem 4.1), and by extension, the rate of disagreement between selective ensembles (Corollary 4.3). In addition, we show that

this also bounds the variance component in the ensembles' bias-variance error decomposition (Domingos, 2000) (Corollary 4.2), providing guidance on how to effectively use of them in practice.

Our experiments show that on seven benchmark datasets, selective ensembles of just ten models either *agree on the entire test data* across random differences in how their constituent models are trained, or abstain at reasonably low rates (1-5% in most cases; Section 5.1). Additionally, we show that simple ensembling doubles the agreement of attributions on key metrics on average, and when the variance of the constituent models is high that selective ensembling further enhances this effect (Section 5.2).

In summary, our contributions are: *(1)* we show that beyond predictions, feature attributions are not consistent across seemingly inconsequential random choices during learning (Section 3); *(2)* we introduce *selective ensembling*, a learning method that *guarantees* bounded inconsistency in predictions, (Section 4); and *(3)* we demonstrate the effectiveness of this approach on seven datasets, showing that selective ensembles consistently predict *all* points across models trained with different random seeds or leave-one-out differences in their training data, while also achieving low abstention rates and higher feature attribution consistency.

## 2    NOTATION AND PRELIMINARIES

We assume a supervised classification setting, with data points $(x,y) \in \mathbf{X} \times \mathbf{Y}$, drawn from data distribution, $\mathcal{D}$, where $x$ represents a vector of features and $y$ a response. In order to capture the effects of arbitrary random events on a learned model—ranging from randomness during training to randomness in the data selection process—we generalize the standard concept of a *learning rule* to that of a *learning pipeline*. Specifically, a learning pipeline, $\mathcal{P}$, is a procedure that outputs a model, $h: \mathbf{X} \to \mathbf{Y}$, taking as input a random state, $S \sim \mathcal{S}$, containing all the information necessary for $\mathcal{P}$ to produce a model (including the architecture, training set, random coin flips used by the learning rule, etc.). Intuitively, $\mathcal{S}$ represents a distribution over random events that might impact the learned model. For example, $\mathcal{S}$ might capture randomness in sampling of the training set, or nondeterminism in the optimization process, e.g., the initialization of parameters, the order in which batches are processed, or the effects of dropout.

In our experiments, we model $\mathcal{S}$ to capture two specific types of random choices, namely *(1)* the initial parameters of the model, and *(2)* leave-one-out changes to the training data. As the initial parameters of the model tend to be determined by a random seed, we will interchangeably refer to this as the selection of random seed. More generally, both of these types of choices instantiate a broader class of choices that could be considered *arbitrary*, despite the fact that they may impact the predictions (Black and Fredrikson, 2021; Marx et al., 2019; Mehrer et al., 2020) (Section 5.1) and explanations (Section 5.2) of the resulting model.

## 3    INSTABILITY OF FEATURE ATTRIBUTIONS IN DEEP MODELS

Before we consider mitigating predictive inconsistency with ensembling, we first demonstrate that models' inconsistency across random choices in training is exhibited not only through its predictions, but through its *feature attributions* as well. Feature attributions refer to numeric scores generated for some set of a model's features—most commonly the model's input features—which are meant to connote how important each feature is in generating the model's prediction. Feature attributions are commonly used as a tool for explaining model behavior (Simonyan et al., 2014; Leino et al., 2018; Sundararajan et al., 2017; Adebayo et al., 2020) localized to given set of inputs. Thus, inconsistent feature attributions between models suggest the models differ in the *process* by which they arrive at their predictions, even if the predictions are the same.

In deep models, many of the most popular attribution methods are based on the model's gradients at or around a given point (Simonyan et al., 2014; Sundararajan et al., 2017). Accordingly, we will focus on the stability of gradients, and show via analysis and experiment that they are not stable in conventional deep models. First, we motivate our results by showing that even two deep models that predict the same labels on all points may have arbitrarily different gradients almost everywhere. Later, in our empirical evaluation (Section 5.2), we demonstrate the extent of the differences between Saliency Maps (Simonyan et al., 2014) (i.e., input gradients) of deep networks even when the randomness of the learning pipeline is controlled to allow only one-point differences in the training set or differences in the random seed.

**Predictions with Arbitrary Gradients.**    We show that even deep models that predict the exact same labels on all points cannot necessarily be expected to have the same, or even similar, gradients; in fact, given a binary classification model $h$, we can construct a model $\hat{h}$ which predicts the same labels as $h$,

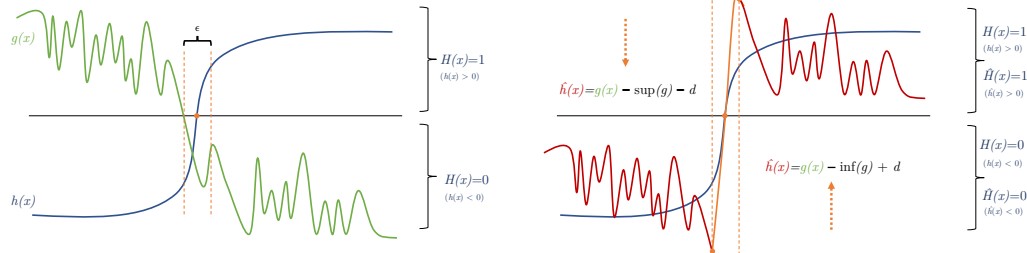

Figure 1: Intuitive illustration of how two models which predict identical classification labels can have arbitrary gradients. To show this, given a binary classifier $H$ and an arbitrary function $g$, we construct a classifier $H'$ that predicts the same labels as $H$, yet has gradients equal to $g$ almost everywhere. We formally state this result in Theorem 3.1.

but has arbitrarily different gradients everywhere except an arbitrarily small region around the boundary of $h$ (Theorem 3.1).

**Theorem 3.1.** *Let $H : \mathbf{X} \to \{-1,1\} = \mathrm{sign}(h)$ be a binary classifier and $g : \mathbb{R}^n \to \mathbb{R}$ be an unrelated function that is bounded from above and below, continuous, and piecewise differentiable. Then there exists another binary classifier $\hat{H} = \mathrm{sign}(\hat{h})$ such that for any $\epsilon > 0$,*

$$\forall x \in \mathbf{X} \, . \qquad \textit{1.} \ \hat{H}(x) = H(x) \qquad \textit{2.} \ \inf_{x' : H(x') \neq H(x)} \left\{ ||x - x'|| \right\} > \epsilon/2 \implies \nabla \hat{h}(x) = \nabla g(x)$$

The proof of Theorem 3.1 is given in Appendix A.1 in the supplementary material. The proof is by construction of $\hat{h}$; a sketch giving the intuition behind the construction is provided in Figure 1. In short, we first partition the domain into contiguous regions that are given the same label by $H$. We then construct $\hat{h}$ from $g$ by adjusting $g$ to lie above or below the origin to match the prediction behavior of $h$ in each region. As these transformations merely shift $g$ by a constant in each region, they do not change $\nabla g$ except near decision boundaries, where it is necessary to move across the origin.

**Observations.** The intuition stemming from Theorem 3.1 is that a model's gradients at each point are largely disconnected from the labels it predicts on a distribution. As models that make identical predictions are likely to have similar loss on a given dataset, this theorem points to the possibility that models of similar objective quality may still have arbitrarily different gradients. In Section 5.2, we demonstrate that this outcome is not only *possible*, but that it occurs in real models—for example, on the German Credit dataset predicting credit risk, on average, individual models with similar accuracy agree on *less than two out of the five most important features* influencing their decision.

## 4 SELECTIVE ENSEMBLING

The results of Section 3 suggests that models that are retrained and redeployed, may exhibit substantially different behavior from their previous iterations. We build on the approach of ensembling for variance reduction by showing how these differences in behavior can be bounded via *selective ensembling*. However, whereas prior work which finds that *more diversity* among the constituent networks is beneficial for reducing overall error (Krogh and Vedelsby, 1995; Hansen and Salamon, 1990; Maclin et al., 1995; Opitz and Shavlik, 1996), our goal is to minimize, or at least place strict bounds on, the variance component. We show that ideas from robust classification, and in particular *randomized smoothing* (Cohen et al., 2019), which stem from recent results on multinomial hypothesis testing (Hung et al., 2019), can be used to enforce such a bound.

**Mode Predictor.** We may view the image of the learning pipeline, $\mathcal{P}$, as a distribution over possible models induced by applying $\mathcal{P}$ to the random state, $S \sim \mathcal{S}$. The *mode prediction* on an input $x$, with respect to $\mathcal{S}$, is the expected label that would be predicted on $x$ by models drawn from this distribution. More formally, we define the *mode predictor*, $g_{\mathcal{P},\mathcal{S}}$ for a pipeline, $\mathcal{P}$, and random state distribution, $\mathcal{S}$, as given by Equation 1.

$$g_{\mathcal{P},\mathcal{S}}(x) = \operatorname*{argmax}_{y \in \mathbf{Y}} \left\{ \mathbb{E}_{S \sim \mathcal{S}} \left[ \mathbb{1}[\mathcal{P}(S\,;x) = y] \right] \right\} \tag{1}$$

**Algorithm 1:** Selective Ensemble Creation

```
def train_ensemble(P, S~Sⁿ, n):
    return {P(Sᵢ) for i∈[n]}

def sample_ensemble(P, S, n):
    S ← sample_iid(Sⁿ)
    return train_ensemble(P, S, n)
```

**Algorithm 2:** Selective Ensemble Prediction

```
def ensemble_predict(ĝₙ(P,S), α, x):
    Y ← Σ_{h∈ĝₙ(P,S)} one_hot(h(x))
    n_A, n_B ← top_2(Y)
    if binom_p_value(n_A, n_A+n_B, 0.5) ≤ α
      then
        return argmax(Y)
    else
        return ABSTAIN
```

Note that while $g_{\mathcal{P},\mathcal{S}}$ is deterministic, and is therefore not sensitive to a specific state drawn from $\mathcal{S}$, it does not necessarily produce the ground truth label on all inputs—some learning pipelines may converge to a stable loss minimum that misclassifies certain points.

**Approximation via Ensembling.** An explicit representation of the true mode predictor is, of course, unattainable—the non-convex loss surface of deep models and the complex interactions between the learning pipeline and the distribution of random states makes the expectation in Equation 1 infeasible to compute analytically. However, we can approximate $g_{\mathcal{P},\mathcal{S}}(x)$ by computing the empirical mode prediction on $x$ over a random sample of models produced by i.i.d. draws from $\mathcal{P}(S)$. But although ensembles with sufficiently many constituent models will more reliably output the mode prediction, for any fixed-size ensemble there will remain points on which the margin of the plurality vote is small enough to "flip" to runner-up in some set of nearby ensembles that differ on a subset of their constituents; in other words, these ensembles will not predict the mode prediction.

To rigorously bound the rate at which the ensemble will differ from the mode prediction, we allow the ensemble to *abstain* on points where the constituent predictions indicate a statistical toss-up between the two most likely classes. We call ensembles that may abstain in this way *selective ensembles*, borrowing the terminology from selective classification (El-Yaniv et al., 2010). We can think of of abstention as a means of flagging unstable points on which the selective ensemble cannot accurately determine the mode prediction; whether this should be interpreted as a failed attempt at classification is an application-specific consideration.

Selective ensembles of $n$ models predict according to the following procedure. First, the predictions of each of the $n$ models in the ensemble are collected. The constituent models are derived from $n$ i.i.d. samples of $\mathcal{P}(S)$ from $\mathcal{S}$, as described in Algorithm 1. From these predictions, we perform a two-sided statistical test to determine if the mode prediction was selected by a statistically significant majority of the constituent models. If the statistical test succeeds, we return the empirical mode prediction; otherwise we abstain from predicting. Pseudocode for this prediction procedure is given in Algorithm 2. We will denote by $\hat{g}_n(\mathcal{P},S)$ (for $S\sim\mathcal{S}^n$) the output of `train_ensemble` in Algorithm 1, and by $\hat{g}_n(\mathcal{P},S\,;\alpha,x)$ prediction produced by `ensemble_predict` in Algorithm 2 on $\hat{g}_n(\mathcal{P},S)$.

Because of their ability to abstain from prediction, we can prove that with probability at least $1-\alpha$, a selective ensemble will either return the true mode prediction or abstain, where $\alpha$ is a chosen threshold for the statistical test to prevent prediction in the case of a toss-up. In other words, on any point on which it does not abstain, a selective ensemble will disagree with the mode predictor, $g_{\mathcal{P},\mathcal{S}}$, with probability at most $\alpha$, as stated formally in Theorem 4.1.

The statement of Theorem 4.1 make use of the relation, $\overset{\text{ABS}}{\neq}$, where $y_1 \overset{\text{ABS}}{\neq} y_2$ if and only if $y_1 \neq$ ABSTAIN and $y_2 \neq$ ABSTAIN and $y_1 \neq y_2$. That is, $\overset{\text{ABS}}{\neq}$ captures disagreement between non-rejected predictions.

**Theorem 4.1.** *Let $\mathcal{P}$ be a learning pipeline, and let $\mathcal{S}$ be a distribution over random states. Further, let $g_{\mathcal{P},\mathcal{S}}$ be the mode predictor, let $\hat{g}_n(\mathcal{P},S)$ for $S\sim\mathcal{S}^n$ be a selective ensemble, and let $\alpha \geq 0$. Then,*

$$\forall x \in \mathbf{X} \ . \ \Pr_{S\sim\mathcal{S}^n}\left[\hat{g}_n(\mathcal{P},S\,;\alpha,x) \overset{\text{ABS}}{\neq} g_{\mathcal{P},\mathcal{S}}(x)\right] \leq \alpha$$

The proof (Appendix A) relies on a result from Hung and Fithian (Hung et al., 2019) which bounds the probability that a set of votes does not return the true plurality outcome, and we apply it in a similar fashion to how it is used for making robust predictions in Randomized Smoothing (Cohen et al., 2019).

Theorem 4.1 states that the probability that a selective ensemble makes a prediction that does not match the mode prediction is small. However, one possible means of ensuring this is by not providing a prediction

Figure 2: The left two plots show abstention rates as a function of the underlying probability of agreement among models over $\mathcal{S}$, i.e., the probability that any given model will return the mode prediction, with plots denoting varying numbers of constituent models. The right two graphs demonstrate the relationship between consistency of the ensemble models as given by Corollary 4.3.

in the first place, i.e., if the selective ensemble abstains. Thus, the *abstention rate* is necessary to quantify the fraction of points on which the mode prediction will actually be produced.

In the 0-1 loss bias-variance decomposition of Domingos (2000), the variance component of a classifier's loss is defined as the expected loss relative to the mode prediction (in our case, taken over the randomness in $\mathcal{S}$). Thus, Theorem 4.1 leads to a direct bound on this component, assuming a bound, $\beta$, on the abstention rate. This is formalized in Corollary 4.2.

**Corollary 4.2.** *Let $\mathcal{P}$ be a learning pipeline, and let $\mathcal{S}$ be a distribution over random states. Further, let $g_{\mathcal{P},\mathcal{S}}$ be the mode predictor, let $\hat{g}_n(\mathcal{P},S)$ for $S \sim \mathcal{S}^n$ be a selective ensemble. Finally, let $\alpha \geq 0$, and let $\beta \geq 0$ be an upper bound on the expected abstention rate of $\hat{g}_n(\mathcal{P},S)$. Then, the expected* loss variance, $V(x)$, over inputs, $x$, is bounded by $\alpha + \beta$. That is,*

$$\mathbb{E}_{x \sim \mathcal{D}}\Big[V(x)\Big] = \mathbb{E}_{x \sim \mathcal{D}}\bigg[\Pr_{S \sim \mathcal{S}^n}\Big[\hat{g}_n(\mathcal{P},S\,;x) \neq g_{\mathcal{P},\mathcal{S}}(x)\Big]\bigg] \leq \alpha + \beta$$

**Consistency of Selective Ensembles.** Using the result from Theorem 4.1, we can also address the original problem raised: that deep models often disagree on their predictions due to arbitrary random events over the training pipeline. We show that, given a bound, $\beta$, on the abstention rate, the probability that two selective ensembles disagree in their predictions is bounded by $2(\alpha + \beta)$ (Corollary 4.3). Intuitively, this suggests that the predictions of selective ensembles are more stable over different instantiations of the random decisions captured by $\mathcal{S}$ compared to individual models.

**Corollary 4.3.** *Let $\mathcal{P}$ be a learning pipeline, and let $\mathcal{S}$ be a distribution over random states. Further, let $\hat{g}_n(\mathcal{P},S)$ for $S \sim \mathcal{S}^n$ be a selective ensemble. Finally, let $\alpha \geq 0$, and let $\beta \geq 0$ be an upper bound on the expected abstention rate of $\hat{g}_n(\mathcal{P},S)$. Then,*

$$\mathbb{E}_{x \sim \mathcal{D}}\bigg[\Pr_{S^1,S^2 \sim \mathcal{S}^n}\Big[\hat{g}_n(\mathcal{P},S^1\,;\alpha,x) \neq \hat{g}_n(\mathcal{P},S^2\,;\alpha,x)\Big]\bigg] \leq 2(\alpha + \beta)$$

Corollary 4.3 tells us that the agreement between any two selective ensembles is at least $1 - 2(\alpha + \beta)$. For a fixed $n$, decreasing $\alpha$ will lead to a higher abstention rate. Thus in order for $\alpha$ *and* $\beta$ to both be small, as would be necessary for a high fraction of consistently-predicted points, we may require a large number of constituent models, $n$. Figure 2 illustrates the trade-off between $\alpha$, $\beta$, and $n$, depending on the base level of agreement of the constituent models. In Section 5, we show empirically that even with small values of $n$, abstention rates of selective ensembles are reasonably low in practice.

In summary, selective ensembles accomplish three primary things: (1) they identify points on which the mode prediction cannot be determined, (2) they bound the fraction of points that can be inconsistently predicted, and (3) they provide a means of reliably inferring the mode prediction when the abstention rate can be kept sufficiently low.

## 5 EVALUATION

In this section, we demonstrate empirically that selective ensembles reduce instability in deep model predictions far below their theoretical bounds—to *zero* inconsistent predictions in the test set over 276

| Randomness | Ger. Credit | Adult | Seizure | Warfarin | Tai. Credit | FMNIST | Colon |
|---|---|---|---|---|---|---|---|
| | | | *mean accuracy $\pm$ standard deviation* | | | | |
| RS | $.730\pm.020$ | $.842\pm1e{-}3$ | $.973\pm2e{-}3$ | $.686\pm3e{-}3$ | $.820\pm1e{-}3$ | $.916\pm3e{-}3$ | $.927\pm2e{-}3$ |
| LOO | $.729\pm.012$ | $.843\pm7e{-}4$ | $.976\pm2e{-}3$ | $.686\pm2e{-}3$ | $.820\pm1e{-}3$ | $.917\pm8e{-}4$ | $.926\pm3e{-}3$ |

Table 1: Mean accuracy over 500 models trained over changes to random initialization and leave-one-out differences in training data. German Credit stands as an outlier due to its small sample size ($|D|{=}800$).

| Randomness | $n$ | Ger. Credit | Adult | Seizure | Tai. Credit | Warfarin | FMNIST | Colon |
|---|---|---|---|---|---|---|---|---|
| | | | | *mean of portion of test data with $p_{flip}>0$* | | | | |
| RS | 1 | .570 | .087 | .060 | .082 | .098 | .061 | .037 |
| RS | (5, 10, 15, 20) | 0.0 | 0.0 | 0.0 | 0.0 | 0.0 | 0.0 | 0.0 |
| LOO | 1 | .262 | .063 | .031 | .031 | .033 | .034 | .042 |
| LOO | (5, 10, 15, 20) | 0.0 | 0.0 | 0.0 | 0.0 | 0.0 | 0.0 | 0.0 |

Table 2: Percentage of points with disagreement between at least one pair of models ($p_{flip}>0$) trained with different random seeds (RS) or leave-one-out differences (LOO) in training data, for single models ($n{=}1$) and selective ensembles ($n{>}1$). Results are averaged over 10 runs of creating 24 selective ensemble models, standard deviations are in Appendix F. Selective ensemble results are together, as there is no disagreement.

| $\mathcal{S}$ | $n$ | Ger. Credit | Adult | Seizure | Warafin | Tai. Credit | FMNIST | Colon |
|---|---|---|---|---|---|---|---|---|
| | | | | *accuracy (abstain as error) / abstention rate* | | | | |
| RS | 5 | 0.0/1.0 | 0.0/1.0 | 0.0/1.0 | 0.0/1.0 | 0.0/1.0 | 0.0/1.0 | 0.0/1.0 |
| RS | 10 | .576/.291 | .820/.043 | .960/.026 | .660/.050 | .800/.039 | .888/.059 | .914/.032 |
| RS | 15 | .636/.205 | .827/.032 | .965/.018 | .668/.037 | .807/.028 | .897/.042 | .919/.023 |
| RS | 20 | .664/.165 | .830/.024 | .967/.014 | .670/.031 | .810/.023 | .902/.036 | .921/.019 |
| LOO | 5 | 0.0/1.0 | 0.0/1.0 | 0.0/1.0 | 0.0/1.0 | 0.0/1.0 | 0.0/1.0 | 0.0/1.0 |
| LOO | 10 | .653/.151 | .827/.032 | .962/.027 | .677/.018 | .812/.017 | .909/.020 | .912/.036 |
| LOO | 15 | .678/.105 | .832/.012 | .968/.019 | .679/.013 | .814/.013 | .910/.016 | .916/.027 |
| LOO | 20 | .689/.079 | .834/.018 | .970/.015 | .680/.011 | .815/.010 | .912/.012 | .919/.023 |

Table 3: Accuracy and abstention rate of selective ensembles, with $n\in\{5,10,15,20\}$ constituents. Results are averaged over 24 randomly selected models; standard deviations are given in Table 8 in Appendix F

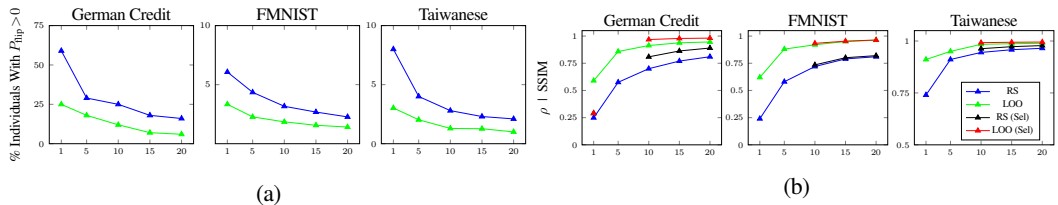

Figure 3: Figure a: Percentage of test data with non-zero disagreement rate in normal (i.e., not selective) ensembles. Horizontal axis depicts ensemble size. Figure b: Average Spearman's Ranking coefficient, $\rho$, (For FMNIST, SSIM) between feature attributions for saliency maps generated for each individual test point (y-axis) over number of ensemble models (x-axis). The lines indicated with (Sel) in the legend are the same metrics for selective ensembles.

pairwise comparisons of model predictions for each of tabular datasets, and 40 for image datasets. Additionally, following Theorem 3.1, we show that feature attributions of individual deep models are frequently inconsistent, and that ensembling effectively mitigates this problem.

**Setup.** To evaluate selective ensembling, we focus on two sources of randomness in the learning rule: *(1)* random initialization, and *(2)* leave-one-out changes to the training set. Our experiments consider seven datasets: UCI German Credit, Adult, Taiwanese Credit Default, Seizure, all from Dua and Karra Taniskidou (2017); the IWPC Warfarin Dosing Recommendation (International Warfarin Pharmacogenetic Consortium, 2009), Fashion MNIST (Xiao et al., 2017), and Colorectal Histology (Kather et al., 2016a). All of these datasets are either related to finance, credit approval, or medical diagnosis, except for FMNIST, which we include as it is a common benchmark for image classification. Further details are in Appendix B.

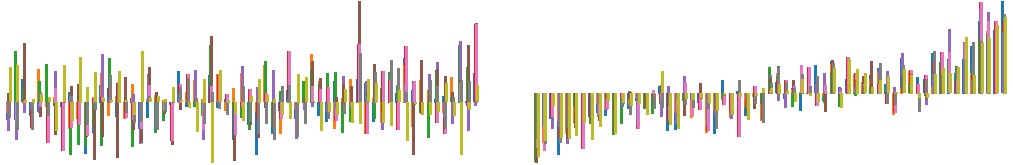

Figure 4: Inconsistency of attributions on the same point across an individual (left) and ensembled (right) model ($n=15$). The height of each bar on the horizontal axis represents the attribution score of a distinct feature, and each color represents a different model. Features are ordered according to the attribution scores of one randomly-selected model.

All experiments are implemented in TensorFlow 2.3. For each tabular, we train $500$ models from independent samples of the relevant source of randomness (e.g. leave-one-out data variations or random seeds), and for each image dataset, we train $200$ models from independent samples of each source of randomness. Details about the model architecture and hyperparameters used are given in Appendix C. Table 1 reports the mean accuracy for each dataset, along with the standard deviation.

For each non-image dataset we generate 24 random ensembles of size $n \in \{5, 10, 15, 20\}$ by selecting uniformly without replacement among the $500$ pre-trained models, as well 24 "singleton" models drawn uniformly from the $500$ to use as a point of comparison when measuring the stability of each ensemble. For image datasets, we generate 10 random ensembles of each size among 200 pre-trained models. We report ensemble predictions in the main paper using $\alpha = 0.05$.

## 5.1 SELECTIVE ENSEMBLES: PREDICTION STABILITY AND ACCURACY

To measure prediction instability over either selective ensembles or singleton models, we compare the predictions of each pair of models on each point in the test set, amounting to 276 comparisons for tabular datasets, and 40 comparisons for image datasets, in total for each point, and record the rate of disagreement, $p_{\text{flip}}$, across these comparisons. We report mean and variance of this disagreement over 10 random re-samplings of constituent models to create ensemble models.

The results in Table 2 and Figure 3a show the percentage of points with disagreement rate greater than zero. We see that for singleton models, as many as 57% of test points have $p_{\text{flip}} > 0$, indicating that disagreement in prediction is in some cases the norm rather than the exception, although more commonly this occurs on 5-10% of the data. *Notably, selective ensembles completely mitigate this effect:* even when *as few as ten* models are included in the ensemble, *no* points experienced $p_{\text{flip}} > 0$. Combined with the fact that abstention rates remain low (1-5%) in all cases except where $p_{\text{flip}}$ was originally very high (e.g., German Credit), this shows that selective ensembling can be a practical method for mitigating prediction instability.

Table 3 shows the accuracy of selective ensembles, with abstention counted towards error, as well as accuracy of non-selective ensembles for comparison. Notably, in all six models, with the exception of German Credit, the abstention rate drops to below $4\%$ with 20 models in the ensemble. Accordingly, the accuracy of the selective ensembles in these cases is comparable—typically within a few points—to that of the traditional ensemble. However, with just five models in the ensemble, the abstention rate is 100%; to achieve reasonable predictions with very few models, the threshold $\alpha$ needs to be increased accordingly. Disagreement of non-selective ensembles are pictured in Figure 3a (with exact numbers in Appendix F): while they do lower prediction inconsistency, they are unable to eliminate it as selective ensembles do.

## 5.2 ATTRIBUTION STABILITY

Following up on the theoretical result given in Theorem 3.1, we demonstrate that feature attributions, which are usually computed for deep models using gradients (Simonyan et al., 2014; Sundararajan et al., 2017; Leino et al., 2018), are often inconsistent between similar models. We then show that, just as ensembling increases prediction stability, it also mitigates gradient instability, leading to more consistent attributions across models. For these experiments, we computed attributions using saliency maps (Simonyan et al., 2014), which are simply the gradient of the model's prediction with respect to its input, as a simple and widely-used representative of gradient-based attribution methods.

| Dataset | Random Seed | | | | Leave-one-out | | | |
|---|---|---|---|---|---|---|---|---|
| | Top-5 | $\rho$ | $r$ | SSIM | Top-5 | $\rho$ | $r$ | SSIM |
| German Credit | 0.20, .27 | 0.01, 0.25 | 0.02, 0.28 | – | 0.20, 0.49 | 0.01, 0.59 | 0.02, 0.60 | – |
| Adult | 0.46, 0.83 | 0.09, 0.83 | 0.07, 0.93 | – | 0.46, 0.89 | 0.15,0.91 | 0.14, 0.95 | – |
| Seizure | 0.14, 0.12 | 0.29, 0.32 | 0.30, 0.33 | – | 0.09, 0.25 | 0.23, 0.57 | 0.24, 0.59 | – |
| Warfarin | 0.37, 0.67 | 0.15, 0.72 | 0.12, 0.73 | – | 0.36, 0.92 | 0.12, 0.96 | 0.11, 0.97 | – |
| Taiwanese Credit | 0.55, 0.76 | 0.35, 0.75 | 0.36,0.83 | – | 0.56,0.91 | 0.35,0.95 | 0.37,0.96 | – |
| FMNIST | 0.00, 0.26 | – | 0.61, 0.61 | 0.50, 0.25 | 0.00, 0.57 | – | 0.90, 0.89 | 0.78, 0.62 |
| Colon | 0.00, 0.63 | – | 0.00, 0.92 | 0.18, 0.82 | 0.00, 0.61 | – | 0.00, 0.91 | 0.18,0.81 |

Table 4: Average top-5 intersection, Spearman's Rank Correlation Coefficient ($\rho$), and Pearson's Correlation Coefficient ($r$) to demonstrate attribution inconsistency on the *same* test points across *different* models. As a baseline, we compare against differences observed on *different* points in the *same* model. The baseline numbers are presented as: similarity baseline, similarity across models. For image models, we also report the Structural Similarity Index (SSIM). Standard deviations are included in Appendix H.2.

**Metrics.** Following previous work (Dombrowski et al., 2019; Ghorbani et al., 2019), we measure the similarity between attributions using Spearman's Ranking Correlation ($\rho$) and the top-$k$ intersection, with $k = 5$. For image datasets, we also display the Structural similarity metric (SSIM), discussed further in Appendix D.1. Spearman's $\rho$ is a natural choice of metric as attributions induce an order of importance among features. We note that the top-$k$ intersection is especially interesting in tabular datasets, as often only the most important features are of explanatory interest. To stay consistent with prior work, we also include Pearson's Correlation Coefficient ($r$). Note that $r$ and $\rho$ vary from -1 to 1, denoting negative, zero, and positive correlation. We compute these metrics over 276 pairwise comparisons of attributions for each size of ensemble (1, 5, 10, 15, and 20) for tabular datasets, and 40 pairwise comparisons for image datasets. For the top-$k$ metric, we report the mean size of the intersection between each pair of attributions. More details are in Appendix D.

**Baselines.** To contextualize the difference of attributions across models trained from distinct randomness, we also include the attribution similarity between 24 randomly chosen points in the *same* model (Table 4). We also present a visual comparison of model attributions, for which we simply plot the attribution for the predicted class for a given point from nine randomly selected models out of the 24, and present the feature attributions in order of their magnitude according to another randomly selected model (Figure 4).

**Singleton Models** The left image in Figure 4 demonstrates the inconsistency of model attributions of individual German Credit models on a random point in the test set. Each bar on the x-axis represents the attributions for a feature, and each different-colored bar represents a different randomly selected model. Thus, the disagreement between the sizes of the bars of different colors shows the disagreement between models on which features should be deemed important. Notably, some of the bars on the graph depicting individual models even have different signs, which means that models disagree on whether that feature counts towards or against the same prediction. Similar graphs for all other datasets are included in Appendix H.1.

We demonstrate this inconsistency further in Table 4. We see that German Credit and Seizure models have particularly unstable attributions, as the top-$k$ (and to a lesser extent, $\rho$ and $r$) scores of attributions of varying points in both the *same* model, and *varying models on the same point*, are quite similar. Feature attributions of individual models are inconsistent even on highly weighted features: e.g., German Credit dataset has a top-$k$ intersection of just over one attribute on average—suggesting that attributions generated through saliency maps on these sets of models may vary substantially over benign retrainings. Even on models where the metrics are higher, e.g. Taiwanese Credit, the baseline similarity between attributions is higher as well—thus, we see that attributions between models of the same point are usually only 2-3$\times$ more related than those of *random points within the same model*.

This instability suggests that salient variables used to inform predictions across models are sensitive to random choices made during training. As previous work has argued in similar contexts (D'Amour et al., 2020), this may be a result of a deep model's under-constrained search space with many local optima equivalent with respect to loss, with several minima corresponding to distinct *rationales* for making predictions.

**Ensemble Models.** We demonstrate that the similarity between saliency maps of ensembled models is greater than that of individual models, and that this similarity increases linearly with the number of models in the ensemble. For these experiments, we average each model's attributions toward the *majority* predicted class of the ensemble. On the right side of Figure 4, we see the feature attributions of various

ensemble models of size 15 over the German Credit dataset. Note how the attributions of ensemble models are much more consistent than on the individual model.

We show this phenomenon more broadly in Figure 3b, where we display graphs of average Spearman's Rank Coefficient ($\rho$) (y-axis) between saliency maps on a point in the test set. We see $\rho$ increase as we increase the number of models in the ensemble (x-axis), for models generated over different random initializations and one-point differences in the training set. Selective ensembles can further increase stability of explanations by abstaining from unstable points, and this has a marked effect when the abstention rate is high (e.g. German Credit). Similar graphs for the rest of metrics calculated are presented in Appendix H.

## 6 RELATED WORK

Prior work has shown that deep models are inconsistent in their predictions across arbitrary random changes in their training pipeline, such as initialization parameters and makeup of the training set (Black and Fredrikson, 2021; Mehrer et al., 2020; D'Amour et al., 2020; Kolen and Pollack, 1991; Feldman, 2019). The problem of model sensitivity, particularly to variability in the training set, can lead to an increase generalization error (Elisseeff et al., 2003) as well as to leaking training set information (Dwork, 2006; Yeom et al., 2018). Thus, stability-enhancing learning rules have received significant attention in order to bolster desirable properties, such as privacy (Liu et al., 2020; Papernot et al., 2018; Wang et al., 2016).

One such approach is model ensembling, which has been used as a variance reduction method since the advent of statistical learning (Zhou et al., 2002; Valentini et al., 2004; Opitz and Maclin, 1999; Tumer and Ghosh, 1996; Dvornik et al., 2019; Hasan et al., 2020; Freund and Schapire, 1997; Sagi and Rokach, 2018; Polikar, 2012; Che et al., 2011; Perrone and Cooper, 1992; Hansen and Salamon, 1990). However, to our knowledge, there is little work on providing guarantees about model disagreement using ensemble models that may *abstain* from prediction. We relate our approach to the classic bias-variance decomposition of error (Domingos, 2000), showing that it certifiably bounds the variance component.

Selective ensembles can be seen as a way to flag points that prone to inconsistency. Under this view, calibration and uncertainty estimation of deep model predictions (Lakshminarayanan et al., 2016; Ovadia et al., 2019) is a related stream of work, and one could potentially use these techniques to determine when to abstain from prediction. However, preventing inconsistent predictions and abstaining from uncertain predictions are different goals: in our setting, the aim is to predict the mode across models drawn from a certain distribution, whereas calibration is measured against predicting the true label. Moreover, prior work has shown that confidence scores may not be correlated with prediction consistency across models with different random initializations (Black and Fredrikson, 2021). Finally, while abstaining on points with low confidence scores may lead to greater consistency, it may not yield a guarantee, which this work provides.

Conformal inference (Linusson et al., 2020; Gupta et al., 2019; Löfström et al., 2013), which rigorously assigns confidence to predictions in settings where the data may differ from training, is similarly related in that such a measure could be useful in identifying inconsistently predicted points. However, in this work, we aim to achieve consistent predictions across a *known* distribution of models, as prior work, as well as our results, suggest, even points conforming to past observations may still be predicted differently by different models.

In addition to inconsistent predictions, this work demonstrates how feature attributions can differ substantially between individual deep models with inconsequential differences. Prior works investigating instability of gradient-based explanation techniques focus on an *adversarial* context (Dombrowski et al., 2019; Ghorbani et al., 2019; Heo et al., 2019; Wang et al., 2020). For example, Anders et al. (2020) develop attacks to create similar models that have differing gradient-based explanations. Contrastingly, this work focuses on the instability of counterfactual explanations between similar models that may occur naturally. As we demonstrate in Section 5.2, model gradients can be quite dissimilar without any adversary.

## 7 CONCLUSION

We show that similar deep models can have not only inconsistent predictions, but substantially different gradients as well. We introduce *selective ensembles* to mitigate this problem by bounding a model's inconsistency over random choices made during training. Empirically, we show that selective ensembles predict *all* points consistently over all datasets we studied. Selective ensembling may present a more reliable way of using deep models in settings where high model complexity *and* stability are required.

## ACKNOWLEDGMENTS

This work was developed with the support of NSF grant CNS-1704845, NSF CNS-1943016, as well as by DARPA and the Air Force Research Laboratory under agreement number FA8750-15-2-0277. The U.S. Government is authorized to reproduce and distribute reprints for Governmental purposes not withstanding any copyright notation thereon. The views, opinions, and/or findings expressed are those of the author(s) and should not be interpreted as representing the official views or policies of DARPA, the Air Force Research Laboratory, the National Science Foundation, or the U.S. Government.

## ETHICS STATEMENT

This work is motivated by the problem of model inconsistency over time in deployment settings—particularly settings impacting individuals' lives, where inconsistency may lead to confusion or even harm to users. The aim of this paper is to prevent harm to those impacted my model decisions occurring from inconsistent model outcomes. We note that in some high-stakes contexts, it is possible that not supplying any outcome (i.e. abstaining) may be worse for an individual than an inconsistent outcome, and so we expect that selective ensembles will be used with a human-in-the-loop or other decision-making framework to adjudicate over abstained-upon points during deployment.

Additionally, recent work has suggested that selective classification can amplify performance disparities between demographic groups (Jones et al., 2020). We investigate the extent of this behavior in selective ensembles and found that by and large, using selective ensembles does not exacerbate accuracy disparity by very much (within 1% of the original disparity), although they did not ameliorate disparities in accuracy that already existed within the performance of the algorithm. The results of these experiments are in Appendix G.

We note that the promise of stability from this paper may encourage machine learning practitioners to over-use highly complex models where a simpler model may be a better choice due to, e.g., transparency requirements. However, we hope that the prospect of increased stability that this paper introduces reduces the harm that can come from machine learning deployment.

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

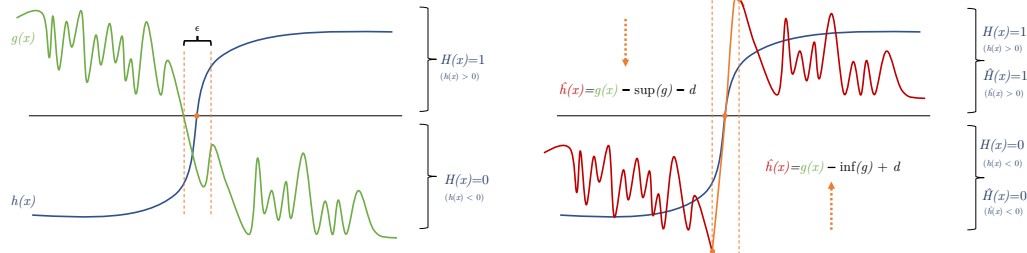

Figure 5: Intuitive illustration of how two models which predict identical classification labels can have arbitrary gradients. To show this, given a binary classifier $H$ and an arbitrary function $g$, we construct a classifier $H'$ that predicts the same labels as $H$, yet has gradients equal to $g$ almost everywhere. We formally state this result in Theorem 3.1.

## A PROOFS

### A.1 PROOF OF THEOREM 3.1

**Theorem 3.1.** *Let $H : \mathbf{X} \to \{-1, 1\} = \mathrm{sign}(h)$ be a binary classifier and $g : \mathbf{X} \to \mathbb{R}$ be an unrelated function that is bounded from above and below, continuous, and piecewise differentiable. Then there exists another binary classifier $\hat{H} = \mathrm{sign}(\hat{h})$ such that for any $\epsilon > 0$,*

$$\forall x \in \mathbf{X}. \qquad 1. \ \hat{H}(x) = H(x) \qquad 2. \ \inf_{x' : H(x') \neq H(x)} \left\{ ||x - x'|| \right\} > \epsilon/2 \implies \nabla \hat{h}(x) = \nabla g(x)$$

*Proof.* We partition $\mathbf{X}$ into regions $\{I_1 \dots I_k\}$ determined by the decision boundaries of $H$. That is, each $I_i$ represents a maximal contiguous region for which each $x \in I_i$ receives the same label from $H$.

Recall we are given a function $g : \mathbf{X} \to \mathbb{R}$ which is bounded from above and below. We create a set of functions $\hat{g}_{I_i} : I_i \to R$ such that

$$\hat{g}_{I_i}(x) = \begin{cases} g(x) - \inf_x g(x) + c & \text{if } H(I_i) = 1 \\ g(x) - \sup_x g(x) - c & \text{if } H(I_i) = -1 \end{cases}$$

where $c$ is some small constant greater than zero. Additionally, let $d(x)$ be the $\ell_2$ distance from $x$ to the nearest decision boundary of $h$, i.e. $d(x) = \inf_{x' : H(x') \neq H(x)} \left\{ ||x - x'|| \right\}$. Then, we define $\hat{h}$ to be:

$$\hat{h}(x) = \begin{cases} \hat{g}_{I_i}(x) & \text{for } x \in I_i \text{ if } d(x) > \frac{\epsilon}{2} \\ \hat{g}_{I_i}(x) \cdot \frac{2d(x)}{\epsilon} & \text{for } x \in I_i \text{ if } d(x) \leq \frac{\epsilon}{2} \end{cases}$$

And, as described above, we define $\hat{H} = \mathrm{sign}(\hat{h})$. First, we show that $\hat{H}(x) = H(x) \ \forall x \in \mathbf{X}$. Without loss of generality, consider some $I_i$ where $H(x) = 1$, for any $x \in I_i$. We first consider the case where $d(x) > \frac{\epsilon}{2}$.

By construction, for $x \in I_i$, $\hat{H}(x) = \mathrm{sign}(\hat{h}(x)) = \mathrm{sign}(\hat{g}_{I_i}(x)) = \mathrm{sign}(g(x) - \inf_x g(x) + c)$. By definition of the infimum, $g(x) - \inf_x g(x) \geq 0$, and thus $\mathrm{sign}(g(x) - \inf_x g(x) + c) = 1$, so $\hat{H}(x) = 1 = H(x)$.

Note that in the case where $d(x) \leq \frac{\epsilon}{2}$, we can follow the same argument as multiplication by a positive constant does not affect the sign. A symmetric argument follows for the case where for $x \in I_i$, $H(x) = -1$; thus, $\hat{H}(x) = H(x) \ \forall x \in \mathbf{X}$.

Secondly, we show that $\nabla \hat{h}(x) = \nabla g(x) \ \forall x$ where $d(x) > \frac{\epsilon}{2}$. Consider the case where $H(x) = 1$. By construction, $\hat{h}(x) = \hat{g}_{I_i}(x) = g(x) - \inf_x g(x) + c$. Note that this means the infimum and $c$ are constants, so their gradients are zero. Thus, $\nabla \hat{h}(x) = \nabla g(x)$. A symmetric argument holds for the case where $H(x) = -1$.

It remains to prove that $\hat{h}$ is continuous and piecewise differentiable, in order to be a realizable as a ReLU-network. By assumption, $g$ is piecewise differentiable, which means that $\hat{g}_i$ are piecewise differentiable as well, as is $\hat{g}_i(x) \cdot \frac{d(x)}{\epsilon}$. Thus, $\hat{h}$ is piecewise-differentiable. To see that $\hat{h}$ is continuous, consider the case where $d(x) = \epsilon/2$ for some $x$. Then $\hat{g}_i(x) \cdot \frac{d(x)}{\epsilon} = \hat{g}_i(x) \cdot \frac{\epsilon}{\epsilon} = \hat{g}_i(x)$. Additionally, consider the case where $d(x) = 0$, i.e. $x$ is on a decision boundary of $h(x)$, between two regions $I_i, I_j$. Then $\hat{h}(x) = \hat{g}_i(x) \cdot \frac{d(x)}{\epsilon} = \hat{g}_i(x) \cdot 0 = 0 = \hat{g}_j(x) \cdot 0 = \hat{g}_j(x)$. This shows that the piecewise components of $\hat{h}$ come to the same value at their intersection. Further, each piecewise component of $\hat{h}$ is equal to some continuous function, as $g(x)$ is continuous by assumption. Thus, $\hat{h}$ is continuous, and we conclude our proof. □

We include a visual intuition of the proof in Figure 5.

## A.2 Proof of Theorem 4.1

**Theorem 4.1.** *Let $\mathcal{P}$ be a learning pipeline, and let $\mathcal{S}$ be a distribution over random states. Further, let $g_{\mathcal{P},\mathcal{S}}$ be the mode predictor, let $\hat{g}_n(\mathcal{P},S)$ for $S \sim \mathcal{S}^n$ be a selective ensemble, and let $\alpha \geq 0$. Then,*

$$\forall x \in \mathbf{X} \ . \ \Pr_{S \sim \mathcal{S}^n}\left[\hat{g}_n(\mathcal{P},S \,;\, \alpha,x) \overset{\text{ABS}}{\neq} g_{\mathcal{P},\mathcal{S}}(x)\right] \leq \alpha$$

*Proof.* $\hat{g}_n(\mathcal{P},S)$ is an ensemble of $n$ models. By the definition of Algorithm 2, $\hat{g}_n(\mathcal{P},S)$ gathers a vector of class counts of the prediction for $x$ from each model in the ensemble. Let the class with the highest count be $c_A$, with counts $n_A$, and the class with the second highest count be called $c_B$, with counts $n_B$. $\hat{g}_n(\mathcal{P},S)$ runs a two-sided hypothesis test to ensure that $\Pr[n_A \sim \text{Binomial}(n_A + n_B, 0.5)] < \alpha$, i.e. that $c_A$ is the true mode prediction over $\mathcal{S}$. See that

$$\Pr\left[g_{\mathcal{P},\mathcal{S}}(x) \neq c_A \ \wedge \ \hat{g}_n(\mathcal{P},S \,;\, \alpha,x) = c_A\right] \tag{2}$$

$$= \Pr\left[g_{\mathcal{P},\mathcal{S}}(x) \neq c_A\right] \cdot \Pr\left[\hat{g}_n(\mathcal{P},S \,;\, \alpha,x) \neq \text{ABSTAIN} \mid g_{\mathcal{P},\mathcal{S}}(x) \neq c_A\right] \tag{3}$$

$$\leq \Pr\left[\hat{g}_n(\mathcal{P},S \,;\, \alpha,x) \neq \text{ABSTAIN} \mid g_{\mathcal{P},\mathcal{S}}(x) \neq c_A\right] \tag{4}$$

$$\leq \Pr\left[\hat{g}_n(\mathcal{P},S \,;\, \alpha,x) \neq \text{ABSTAIN} \mid g_{\mathcal{P},\mathcal{S}}(x) \neq c_A\right] = \alpha \qquad \text{By Hung et al. (2019)} \tag{5}$$

Thus,

$$\Pr\left[g_{\mathcal{P},\mathcal{S}}(x) \neq c_A \ \wedge \ \hat{g}_n(\mathcal{P},S \,;\, \alpha,x) = c_A\right] \leq \alpha$$

□

## A.3 Proof of Corollary 4.2

**Corollary 4.2.** *Let $\mathcal{P}$ be a learning pipeline, and let $\mathcal{S}$ be a distribution over random states. Further, let $g_{\mathcal{P},\mathcal{S}}$ be the mode predictor, let $\hat{g}_n(\mathcal{P},S)$ for $S \sim \mathcal{S}^n$ be a selective ensemble. Finally, let $\alpha \geq 0$, and let $\beta \geq 0$ be an upper bound on the expected abstention rate of $\hat{g}_n(\mathcal{P},S)$. Then, the expected loss variance, $V(x)$, over inputs, $x$, is bounded by $\alpha + \beta$. That is,*

$$\mathbb{E}_{x \sim \mathcal{D}}\left[V(x)\right] = \mathbb{E}_{x \sim \mathcal{D}}\left[\Pr_{S \sim \mathcal{S}^n}\left[\hat{g}_n(\mathcal{P},S \,;\, x) \neq g_{\mathcal{P},\mathcal{S}}(x)\right]\right] \leq \alpha + \beta$$

*Proof.* Since $g_{\mathcal{P},\mathcal{S}}$ never abstains, we have by the law of total probability that

$$\Pr_{S \sim \mathcal{S}^n}\left[\hat{g}_n(\mathcal{P},S \,;\, \alpha,x) \neq g_{\mathcal{P},\mathcal{S}}(x)\right] = \Pr_{S \sim \mathcal{S}^n}\left[\hat{g}_n(\mathcal{P},S \,;\, \alpha,x) \overset{\text{ABS}}{\neq} g_{\mathcal{P},\mathcal{S}}(x) \ \vee \ \hat{g}_n(\mathcal{P},S \,;\, \alpha,x) = \text{ABSTAIN}\right]$$

$$= \Pr_{S \sim \mathcal{S}^n}\left[\hat{g}_n(\mathcal{P},S \,;\, \alpha,x) \overset{\text{ABS}}{\neq} g_{\mathcal{P},\mathcal{S}}(x)\right] + \Pr_{S \sim \mathcal{S}^n}\left[\hat{g}_n(\mathcal{P},S \,;\, \alpha,x) = \text{ABSTAIN}\right]$$

By Theorem 4.1, we have that $\Pr_{S \sim \mathcal{S}^n}\left[\hat{g}_n(\mathcal{P},S \,;\, \alpha,x) \overset{\text{\tiny ABS}}{\neq} g_{\mathcal{P},\mathcal{S}}(x)\right] \leq \alpha$, thus

$$\mathbb{E}_{x \sim \mathcal{D}}\left[\Pr_{S \sim \mathcal{S}^n}\left[\hat{g}_n(\mathcal{P},S \,;\, \alpha,x) \neq g_{\mathcal{P},\mathcal{S}}(x)\right]\right] \leq \alpha + \mathbb{E}_{x \sim \mathcal{D}}\left[\Pr_{S \sim \mathcal{S}^n}\left[\hat{g}_n(\mathcal{P},S \,;\, \alpha,x) = \texttt{ABSTAIN}\right]\right]$$

Finally, since $\beta$ is an upper bound on the expected abstention rate of $\hat{g}_n(\mathcal{P},S)$, we conclude that

$$\mathbb{E}_{x \sim \mathcal{D}}\left[\Pr_{S \sim \mathcal{S}^n}\left[\hat{g}_n(\mathcal{P},S \,;\, \alpha,x) \neq g_{\mathcal{P},\mathcal{S}}(x)\right]\right] \leq \alpha + \beta$$

$\square$

### A.4 PROOF OF COROLLARY 4.3

**Corollary 4.3.** *Let $\mathcal{P}$ be a learning pipeline, and let $\mathcal{S}$ be a distribution over random states. Further, let $\hat{g}_n(\mathcal{P},S)$ for $S \sim \mathcal{S}^n$ be a selective ensemble. Finally, let $\alpha \geq 0$, and let $\beta \geq 0$ be an upper bound on the expected abstention rate of $\hat{g}_n(\mathcal{P},S)$. Then,*

$$\mathbb{E}_{x \sim \mathcal{D}}\left[\Pr_{S^1,S^2 \sim \mathcal{S}^n}\left[\hat{g}_n(\mathcal{P},S^1 \,;\, \alpha,x) \neq \hat{g}_n(\mathcal{P},S^2 \,;\, \alpha,x)\right]\right] \leq 2(\alpha + \beta)$$

*Proof.* For $i \in \{1,2\}$, let $A^i$ be the event that $\hat{g}_n(\mathcal{P},S^i \,;\, \alpha,x) = \texttt{ABSTAIN}$, and let $N^i$ be the event that $\hat{g}_n(\mathcal{P},S^i \,;\, \alpha,x) \overset{\text{\tiny ABS}}{\neq} g_{\mathcal{P},\mathcal{S}}$. In the worst case, $A^1$ and $A^2$, and $N^1$ and $N^2$ are disjoint, that is, e.g., if $\hat{g}_n(\mathcal{P},S^i)$ abstains on $x$, then $\hat{g}_n(\mathcal{P},S^1 \,;\, \alpha,x) \neq \hat{g}_n(\mathcal{P},S^2 \,;\, \alpha,x)$. In other words, we have that

$$\Pr_{S^1,S^2 \sim \mathcal{S}^n}\left[\hat{g}_n(\mathcal{P},S^1 \,;\, \alpha,x) \neq \hat{g}_n(\mathcal{P},S^2 \,;\, \alpha,x)\right] \leq \Pr\left[A^1 \vee A^2 \vee N^1 \vee N^2\right]$$

which, by union bound implies that

$$\Pr_{S^1,S^2 \sim \mathcal{S}^n}\left[\hat{g}_n(\mathcal{P},S^1 \,;\, \alpha,x) \neq \hat{g}_n(\mathcal{P},S^2 \,;\, \alpha,x)\right] \leq \Pr\left[A^1\right] + \Pr\left[A^2\right] + \Pr\left[N^1\right] + \Pr\left[N^2\right].$$

By Theorem 4.1 $\Pr\left[N^i\right] \leq \alpha$. Thus we have

$$\mathbb{E}_{x \sim \mathcal{D}}\left[\Pr_{S^1,S^2 \sim \mathcal{S}^n}\left[\hat{g}_n(\mathcal{P},S^1 \,;\, \alpha,x) \neq \hat{g}_n(\mathcal{P},S^2 \,;\, \alpha,x)\right]\right] \leq 2\alpha + \mathbb{E}_{x \sim \mathcal{D}}\left[\Pr\left[A^1\right]\right] + \mathbb{E}_{x \sim \mathcal{D}}\left[\Pr\left[A^2\right]\right].$$

Finally, since $\beta$ is an upper bound on the expected abstention rate of $\hat{g}_n(\mathcal{P},S)$, we conclude that

$$\mathbb{E}_{x \sim \mathcal{D}}\left[\Pr_{S^1,S^2 \sim \mathcal{S}^n}\left[\hat{g}_n(\mathcal{P},S^1 \,;\, \alpha,x) \neq \hat{g}_n(\mathcal{P},S^2 \,;\, \alpha,x)\right]\right] \leq 2(\alpha + \beta)$$

$\square$

## B DATASETS

The German Credit and Taiwanese data sets consist of individuals financial data, with a binary response indicating their creditworthiness. For the German Credit dataset, there are 1000 points, and 20 attributes. We one-hot encode the data to get 61 features, and standardize the data to zero mean and unit variance using SKLearn Standard scaler. We partitioned the data intro a training set of 700 and a test set of 200. The Taiwanese credit dataset has 30,000 instances with 24 attributes. We one-hot encode the data to get 32 features and normalize the data to be between zero and one. We partitioned the data intro a training set of 22500 and a test set of 7500.

The Adult dataset consists of a subset of publicly-available US Census data, binary response indicating annual income of $> 50k$. There are 14 attributes, which we one-hot encode to get 96 features. We normalize the numerical features to have values between 0 and 1. After removing instances with missing

values, there are 30,162 examples which we split into a training set of 14891, a leave one out set of 100, and a test set of 1501 examples.

The Seizure dataset comprises time-series EEG recordings for 500 individuals, with a binary response indicating the occurrence of a seizure. This is represented as 11500 rows with 178 features each. We split this into 7,950 train points and 3,550 test points. We standardize the numeric features to zero mean and unit variance.

The Warfain dataset is collected by the International Warfarin Pharmacogenetics Consortium (International Warfarin Pharmacogenetic Consortium, 2009) about patients who were prescribed warfarin. We removed rows with missing values, 4819 patients remained in the dataset. The inputs to the model are demographic (age, height, weight, race), medical (use of amiodarone, use of enzyme inducer), and genetic (VKORC1, CYP2C9) attributes. Age, height, and weight are real-valued and were scaled to zero mean and unit variance. The medical attributes take binary values, and the remaining attributes were one-hot encoded. The output is the weekly dose of warfarin in milligrams, which we encode as "low", "medium", or "high", following the recommendations set by International Warfarin Pharmacogenetic Consortium (2009).

Fashion MNIST contains images of clothing items, with a multilabel response of 10 classes. There are 60000 training examples and 10000 test examples. We pre-process the data by normalizing the numerical values in the image array to be between $0$ and $1$.

The colorectal histology dataset contains images of human colorectal cancer, with a multilabel response of 8 classes. There are 5,000 images, which we divide into a training set of 3750 and a validation set of 1250. We pre-process the data by normalizing the numerical values in the image array to be between $0$ and $1$.

The UCI datasets as well as FMNIST are under an MIT license, the colorectal histology and Warfarin datasets are under a Creative Commons License. (Dua and Karra Taniskidou, 2017; Kather et al., 2016b; International Warfarin Pharmacogenetic Consortium, 2009).

## C  MODEL ARCHITECTURE AND HYPER-PARAMETERS

The German Credit and Seizure models have three hidden layers, of size 128, 64, and 16. Models on the Adult dataset have one hidden layer of 200 neurons. Models on the Taiwanese dataset have two hidden layers of 32 and 16. The Warfarin models have one hidden layer of 100. The FMNIST model is a modified LeNet architecture (LeCun et al., 1995). This model is trained with dropout. The Colon models are trained with a modified, ResNet50 (He et al., 2016), pre-trained on ImageNet (Deng et al., 2009), available from Keras. German Credit, Adult, Seizure, Taiwanese, and Warfarin models are trained for 100 epochs; FMNIST for 50, and Colon models are trained for 20 epochs. German Credit models are trained with a batch size of 32; FMNIST 64; Adult, Seizure, and Warfarin models with batch sizes of 128; and Colon and Taiwanese Credit models with batch sizes of 512. German Credit, Adult, Seizure, Taiwanese Credit, Warfarin, and Colon are trained with keras' Adam optimizer with the default parameters. FMNIST models are trained with keras' SGD optimizer with the default parameters.

Note that we discuss train-test splits and data preprocessing above in Section B. We prepare different models for the same dataset using Tensorflow 2.3.0 and all computations are done using a Titan RTX accelerator on a machine with 64 gigabytes of memory.

## D  METRICS

We report similarity between feature attributions with Spearman's Ranking Correlation ($\rho$), Pearson's Correlation Coefficient ($r$), top-$k$ intersection, $\ell_2$ distance, and SSIM for image datasets. We use standard implementations for Spearman's Ranking Correlation ($\rho$) and Pearson's Correlation Coefficient ($r$) from scipy, and implement $\ell_2$ distance as well as the top-$k$ using numpy functions.

Note that $r$ and $\rho$ vary from -1 to 1, denoting negative, zero, and positive correlation. We display top-$k$ for $k$=5, and compute this by taking the number of features in the intersection of the top $5$ between two models, and then diving this by $5$. Thus top-$k$ is between 0 and 1, indicating low and high correlation respectively.

The $\ell_2$ distance has a minimum of $0$, but is unbounded from above, and SSIM varies from -1 to 1, indicating no correlation to exact correlation respectively.

| | | mean $\pm$ std. dev of portion of test data with $p_{\text{flip}} > 0$ | | | | | | |
|---|---|---|---|---|---|---|---|---|
| Randomness | $n$ | Ger. Credit | Adult | Seizure | Tai. Credit | Warfarin | FMNIST | Colon |
| RS | 1 | .570±.020 | .087±.001 | .060±.01 | .082±.002 | .098±.003 | .061±.008 | .037±.005 |
| RS | (5, 10, 15, 20) | 0.0±0.0 | 0.0±0.0 | 0.0±0.0 | 0.0±0.0 | 0.0±0.0 | 0.0±0.0 | 0.0±0.0 |
| LOO | 1 | .262±.014 | .063±.001 | .031±.001 | .031±.001 | .033±.003 | .034±.004 | .042±.005 |
| LOO | (5, 10, 15, 20) | 0.0±0.0 | 0.0±0.0 | 0.0±0.0 | 0.0±0.0 | 0.0±0.0 | 0.0±0.0 | 0.0±0.0 |

Table 5: The percentage of points with disagreement between at least one pair of models ($p_{\text{flip}} > 0$) trained with different random seeds (RS) or leave-one-out differences in training data, for singleton models ($n = 1$) and selective ensembles ($n > 1$). Results for selective ensembles all selective ensembles are shown together, as they all have no disagreement. Note that these results are for $\alpha = 0.01$. But this different $\alpha$ also leads to zero disagreement between predicted points.

Note that we compute these metrics between two different models on the same point, for every point in the test set, over 276 different pairs of models for tabular datasets and over 40 pairs of models for image datasets. We average this result over the points in the test set and over the comparisons to get the numbers displayed in the tables and graphs throughout the paper.

## D.1 SSIM

Explanations for image models can be interpreted as an image (as there is an attribution for each pixel), and are often evaluated visually (Leino et al., 2018; Simonyan et al., 2014; Sundararajan et al., 2017). However, pixel-wise indicators for similarity between images (such as top-k similarity between pixel values, Spearman's ranking coefficient, or mean squared error) often do not capture how similar images are visually, in aggregate. In order to give an indication if the entire explanation for an image model, i.e. the explanatory image produced, is similar, we use the structural similarity index (SSIM) (Wang et al., 2004). We use the implementation from $\mathtt{scikit-image}$ (str). SSIM varies from -1 to 1, indicating no correlation to exact correlation respectively.

## E  EXPERIMENTAL RESULTS FOR $\alpha = 0.01$

We include results on the prediction of selective ensemble models for $\alpha = 0.01$ as well. We include the percentage of points with disagreement between at least one pair of models ($p_{\text{flip}} > 0$) trained with different random seeds (RS) or leave-one-out differences in training data, for singleton models ($n = 1$) and selective ensembles ($n > 1$) in Table 5. Notice the number of points with $p_{\text{flip}} > 0$ is again zero. We also include the mean and standard deviation of accuracy and abstention rate for $\alpha = 0.01$ in Table 6.

## F  SELECTIVE ENSEMBLING FULL RESULTS

We include the full results from the evaluation section, including error bars on the disagreement, accuracy, abstention rates of selective ensembles, in Table 7 and Table 8 respectively. We also include the results for all datasets on the accuracy of non-selective ensembling and their ability to mitigate disagreement, in Table 7 and Table 6 respectively.

## G  SELECTIVE ENSEMBLES AND DISPARITY IN SELECTIVE PREDICTION

In light of the fact that prior work has brought to light the possibility of selective prediction exacerbating model accuracy disparity between demographic groups Jones et al. (2020), we present the selective ensemble accuracy and abstention rate group-by-group for several different demographic groups across four datasets: Adult, German Credit, Taiwanese Credit, and Warfarin Dosing. Results are in Table 9.

## H  EXPLANATION CONSISTENCY FULL RESULTS

We give full results for selective and non-selective ensembling's mitigation of inconsistency in feature attributions.

| $\mathcal{S}$ | $n$ | Ger. Credit | Adult | Seizure | Wafarin | Tai. Credit | FMNIST | Colon |
|---|---|---|---|---|---|---|---|---|
| | | | | *mean accuracy (abstain as error) / std. dev* | | | | |
| RS | 5 | $0.0\pm0.0$ | $0.0\pm0.0$ | $0.0\pm0.0$ | $0.0\pm0.0$ | $0.0\pm0.0$ | $0.0\pm0.0$ | $0.0\pm0.0$ |
| RS | 10 | $.461\pm.016$ | $.807\pm1e{-}3$ | $.945\pm2e{-}3$ | $.646\pm3e{-}3$ | $.788\pm2e{-}3$ | $.870\pm5e{-}3$ | $.902\pm2e{-}3$ |
| RS | 15 | $.589\pm.015$ | $.822\pm8e{-}4$ | $.961\pm1e{-}3$ | $.661\pm3e{-}3$ | $.802\pm9e{-}4$ | $.890\pm2e{-}3$ | $.915\pm1e{-}3$ |
| RS | 20 | $.593\pm.011$ | $.822\pm7e{-}4$ | $.961\pm8e{-}4$ | $.662\pm1e{-}3$ | $.803\pm9e{-}4$ | $.991\pm1e{-}3$ | $.926\pm1e{-}3$ |
| LOO | 5 | $0.0\pm0.0$ | $0.0\pm0.0$ | $0.0\pm0.0$ | $0.0\pm0.0$ | $0.0\pm0.0$ | $0.0\pm0.0$ | $0.0\pm0.0$ |
| LOO | 10 | $.618\pm.017$ | $.818\pm1e{-}3$ | $.947\pm4e{-}3$ | $.674\pm2e{-}3$ | $.807\pm1e{-}3$ | $.904\pm6e{-}4$ | $.901\pm2e{-}3$ |
| LOO | 15 | $.656\pm.017$ | $.828\pm1e{-}3$ | $.963\pm1e{-}3$ | $.678\pm9e{-}4$ | $.812\pm9e{-}4$ | $.908\pm1e{-}3$ | $.912\pm2e{-}3$ |
| LOO | 20 | $.661\pm.018$ | $.829\pm7e{-}4$ | $.964\pm1e{-}3$ | $.678\pm7e{-}4$ | $.812\pm8e{-}4$ | $.909\pm6e{-}4$ | $.912\pm2e{-}3$ |

| $\mathcal{S}$ | $n$ | Ger. Credit | Adult | Seizure | Warfarin | Tai. Credit | FMNIST | Colon |
|---|---|---|---|---|---|---|---|---|
| | | | | *mean abstention rate / std dev* | | | | |
| RS | 5 | $1.0\pm0.0$ | $1.0\pm0.0$ | $1.0\pm0.0$ | $1.0\pm0.0$ | $1.0\pm0.0$ | $1.0\pm0.0$ | $1.0\pm0.0$ |
| RS | 10 | $.449\pm.021$ | $.068\pm2e{-}3$ | $.045\pm2e{-}3$ | $.078\pm5e{-}3$ | $.063\pm2e{-}3$ | $.087\pm8e{-}3$ | $.050\pm3e{-}3$ |
| RS | 15 | $.278\pm.017$ | $.041\pm1e{-}3$ | $.025\pm1e{-}3$ | $.049\pm3e{-}3$ | $.037\pm1e{-}3$ | $.055\pm2e{-}3$ | $.030\pm2e{-}3$ |
| RS | 20 | $.270\pm.015$ | $.040\pm1{-}e3$ | $.024\pm1e{-}3$ | $.047\pm2e{-}3$ | $.036\pm1e{-}3$ | $.054\pm9e{-}4$ | $.038\pm1e{-}3$ |
| LOO | 5 | $1.0\pm0.0$ | $1.0\pm0.0$ | $1.0\pm0.0$ | $1.0\pm0.0$ | $1.0\pm0.0$ | $1.0\pm0.0$ | $1.0\pm0.0$ |
| LOO | 10 | $.215\pm.030$ | $.049\pm2e{-}3$ | $.045\pm5e{-}3$ | $.027\pm2e{-}3$ | $.025\pm1e{-}3$ | $.029\pm1e{-}3$ | $.054\pm2e{-}3$ |
| LOO | 15 | $.144\pm0.040$ | $.030\pm2e{-}3$ | $.026\pm1e{-}3$ | $.017\pm2e{-}3$ | $.017\pm2e{-}3$ | $.021\pm3e{-}3$ | $.035\pm2e{-}3$ |
| LOO | 20 | $.135\pm.040$ | $.029\pm1e{-}3$ | $.025\pm1e{-}3$ | $.017\pm1e{-}3$ | $.017\pm2e{-}3$ | $.019\pm1e{-}3$ | $.035\pm3e{-}3$ |

Table 6: Accuracy (above) and abstention rate (below) of selective ensembles with $n \in \{5,10,15,20\}$ constituents. Results are averaged over 24 models, standard deviation is presented. Note that these results are for $\alpha = 0.01$.

| Randomness | $n$ | Ger. Credit | Adult | Seizure | Tai. Credit | Warfarin | FMNIST | Colon |
|---|---|---|---|---|---|---|---|---|
| | | | | *mean $\pm$ std. dev of portion of test data with $p_{\text{flip}} > 0$* | | | | |
| RS | 1 | $.570\pm.020$ | $.087\pm.001$ | $.060\pm.01$ | $.082\pm.002$ | $.098\pm.003$ | $.061\pm.008$ | $.037\pm.005$ |
| RS | (5, 10, 15, 20) | $0.0\pm0.0$ | $0.0\pm0.0$ | $0.0\pm0.0$ | $0.0\pm0.0$ | $0.0\pm0.0$ | $0.0\pm0.0$ | $0.0\pm0.0$ |
| LOO | 1 | $.262\pm.014$ | $.063\pm.001$ | $.031\pm.001$ | $.031\pm.001$ | $.033\pm.003$ | $.034\pm.004$ | $.042\pm.005$ |
| LOO | (5, 10, 15, 20) | $0.0\pm0.0$ | $0.0\pm0.0$ | $0.0\pm0.0$ | $0.0\pm0.0$ | $0.0\pm0.0$ | $0.0\pm0.0$ | $0.0\pm0.0$ |

Table 7: Percentage of points with disagreement between at least one pair of models ($p_{\text{flip}} > 0$) trained with different random seeds (RS) or leave-one-out differences in training data, for singleton models ($n=1$) and selective ensembles ($n>1$). We present the mean and standard deviation of this percentage over 10 runs of re-sampling ensemble models. Note that these results are for $\alpha = 0.05$ **and** $\alpha = 0.01$, since both resulted in zero inconsistent prediction over predicted points.

| $\mathcal{S}$ | $n$ | Ger. Credit | Adult | Seizure | Warfarin | Tai. Credit | FMNIST | Colon |
|---|---|---|---|---|---|---|---|---|
| | | | | *mean accuracy (abstain as error) / std. dev* | | | | |
| RS | 5 | $0.0\pm0.0$ | $0.0\pm0.0$ | $0.0\pm0.0$ | $0.0\pm0.0$ | $0.0\pm0.0$ | $0.0\pm0.0$ | $0.0\pm0.0$ |
| RS | 10 | $.576\pm.013$ | $.820\pm8e{-}4$ | $.960\pm1e{-}3$ | $.660\pm2e{-}3$ | $.800\pm1e{-}3$ | $.888\pm2e{-}3$ | $.914\pm1e{-}3$ |
| RS | 15 | $.636\pm.017$ | $.827\pm5e{-}4$ | $.965\pm1e{-}3$ | $.668\pm2e{-}3$ | $.807\pm9e{-}4$ | $.897\pm2e{-}3$ | $.919\pm1e{-}3$ |
| RS | 20 | $.664\pm.014$ | $.830\pm5e{-}4$ | $.967\pm9e{-}4$ | $.670\pm3e{-}3$ | $.810\pm8e{-}4$ | $.902\pm1e{-}3$ | $.921\pm1e{-}3$ |
| LOO | 5 | $0.0\pm0.0$ | $0.0\pm0.0$ | $0.0\pm0.0$ | $0.0\pm0.0$ | $0.0\pm0.0$ | $0.0\pm0.0$ | $0.0\pm0.0$ |
| LOO | 10 | $.653\pm.017$ | $.827\pm1e{-}3$ | $.962\pm2e{-}3$ | $.677\pm1e{-}3$ | $.812\pm1e{-}3$ | $.909\pm4e{-}4$ | $.912\pm1e{-}3$ |
| LOO | 15 | $.678\pm.014$ | $.832\pm7e{-}4$ | $.968\pm9e{-}4$ | $.679\pm9e{-}4$ | $.814\pm9e{-}4$ | $.910\pm1e{-}3$ | $.916\pm2e{-}3$ |
| LOO | 20 | $.689\pm.014$ | $.834\pm7e{-}4$ | $.970\pm1e{-}3$ | $.680\pm7e{-}4$ | $.815\pm8e{-}4$ | $.911\pm4e{-}4$ | $.918\pm8e{-}4$ |

| $\mathcal{S}$ | $n$ | Ger. Credit | Adult | Seizure | Warfarin | Tai. Credit | FMNIST | Colon |
|---|---|---|---|---|---|---|---|---|
| | | | | *mean abstention rate / std dev* | | | | |
| RS | 5 | $1.0\pm0.0$ | $1.0\pm0.0$ | $1.0\pm0.0$ | $1.0\pm0.0$ | $1.0\pm0.0$ | $1.0\pm0.0$ | $1.0\pm0.0$ |
| RS | 10 | $.291\pm.014$ | $.043\pm1e{-}3$ | $.02\pm1e{-}3$ | $.050\pm3e{-}3$ | $.039\pm2e{-}3$ | $.059\pm2e{-}3$ | $.032\pm3e{-}3$ |
| RS | 15 | $.205\pm.020$ | $.032\pm1e{-}3$ | $.018\pm1e{-}3$ | $.037\pm3e{-}3$ | $.028\pm1e{-}3$ | $.042\pm2e{-}3$ | $.023\pm2e{-}3$ |
| RS | 20 | $.165\pm.015$ | $.024\pm7{-}e4$ | $.014\pm7e{-}4$ | $.031\pm4e{-}3$ | $.023\pm8e{-}4$ | $.036\pm1e{-}3$ | $.019\pm2e{-}3$ |
| LOO | 5 | $1.0\pm0.0$ | $1.0\pm0.0$ | $1.0\pm0.0$ | $1.0\pm0.0$ | $1.0\pm0.0$ | $1.0\pm0.0$ | $1.0\pm0.0$ |
| LOO | 10 | $.151\pm.041$ | $.032\pm2e{-}3$ | $.027\pm2e{-}3$ | $.018\pm2e{-}3$ | $.017\pm2e{-}3$ | $.020\pm5e{-}4$ | $.036\pm3e{-}3$ |
| LOO | 15 | $.105\pm0.034$ | $.022\pm1e{-}3$ | $.019\pm1e{-}3$ | $.013\pm2e{-}3$ | $.013\pm2e{-}3$ | $.016\pm2e{-}3$ | $.027\pm2e{-}3$ |
| LOO | 20 | $.079\pm.029$ | $.018\pm1e{-}3$ | $.015\pm1e{-}3$ | $.011\pm2e{-}3$ | $.010\pm1e{-}3$ | $.012\pm8e{-}4$ | $.023\pm2e{-}3$ |

Table 8: Accuracy (above) and abstention rate (below) of selective ensembles with $n \in \{5,10,15,20\}$ constituents. Results are averaged over 24 models, standard deviation is presented. Note that these results are for $\alpha = 0.05$, which are presented in the main paper.

*disagreement of non-abstaining ensembles*

| $\mathcal{S}$ | $n$ | Ger. Credit | Adult | Seizure | Tai. Credit | Warfarin | FMNIST | Colon |
|---|---|---|---|---|---|---|---|---|
| RS | 1 | .570±.020 | .087±.001 | .060±.01 | .082±.002 | .098±.003 | 0.113±.005 | .066±.002 |
| RS | 5 | .305±.017 | .045±.001 | .028±.001 | .082±.002 | .054±.003 | .046±.002 | .022±.001 |
| RS | 10 | .234±.014 | .031±.001 | .019±.001 | .041±.001 | .040±.002 | .032±.002 | .014±.002 |
| RS | 15 | .185±.012 | .026±.001 | .015±.001 | .030±.000 | .033±.002 | .028±.002 | .012±.001 |
| RS | 20 | .155±.010 | .022±.001 | .013±.001 | .021±.001 | .030±.002 | .026±.001 | .010±.001 |
| LOO | 1 | .262±.014 | .063±.001 | .031±.001 | .031±.001 | .033±.003 | .056±.004 | .068±.003 |
| LOO | 5 | .142±.037 | .033±.001 | .028±.001 | .019±.001 | .018±.001 | .032±.002 | .030±.003 |
| LOO | 10 | .111±.020 | .023±.001 | .020±.001 | .014±.001 | .016±.001 | .034±.002 | .016±.003 |
| LOO | 15 | .074±.020 | .019±.001 | .017±.001 | .011±.001 | .012±.001 | .029±.001 | .014±.002 |
| LOO | 20 | .067±.013 | .016±.001 | .015±.001 | .010±.000 | .011±.001 | .027±.001 | .010±.001 |

Figure 6: Mean and standard deviation of the percentage of test data with non-zero disagreement over 24 normal (i.e., not selective) ensembles. The mean and standard deviation are taken over ten re-samplings of 24 ensembles. While ensembling alone mitigates much of the prediction instability, it is unable to eliminate it as selective ensembles do.

*accuracy of non-abstaining ensembles*

| $\mathcal{S}$ | $n$ | Ger. Credit | Adult | Seizure | Warfarin | Tai. Credit | FMNIST | Colon |
|---|---|---|---|---|---|---|---|---|
| RS | 5 | 0.745±0.013 | 0.842±0.001 | 0.975±0.001 | 0.688±0.0 | 0.822±0.001 | 0.919±0.001 | 0.927±0.001 |
| RS | 10 | 0.747±0.014 | 0.843±0.001 | 0.975±0.001 | 0.688±0.0 | 0.822±0.001 | 0.92±0.001 | 0.928±0.001 |
| RS | 15 | 0.75±0.01 | 0.842±0.001 | 0.975±0.001 | 0.688±0.0 | 0.822±0.001 | 0.92±0.001 | 0.928±0.001 |
| RS | 20 | 0.747±0.01 | 0.842±0.0 | 0.975±0.001 | 0.688±0.0 | 0.822±0.001 | 0.92±0.001 | 0.928±0.0 |
| LOO | 5 | 0.728±0.011 | 0.844±0.0 | 0.979±0.001 | 0.685±0.002 | 0.821±0.001 | 0.918±0.0 | 0.927±0.002 |
| LOO | 10 | 0.728±0.008 | 0.844±0.001 | 0.978±0.001 | 0.686±0.002 | 0.821±0.001 | 0.918±0.0 | 0.927±0.002 |
| LOO | 15 | 0.733±0.008 | 0.844±0.0 | 0.979±0.001 | 0.685±0.001 | 0.821±0.0 | 0.917±0.0 | 0.927±0.001 |
| LOO | 20 | 0.73±0.008 | 0.843±0.0 | 0.979±0.001 | 0.685±0.002 | 0.821±0.0 | 0.918±0.001 | 0.927±0.001 |

Figure 7: Accuracy of non-selective (regular) ensembles with $n \in \{5,10,15,20\}$ constituents. Results are averaged over 24 models, standard deviation is presented.

## H.1 ATTRIBUTIONS

We pictorially show the inconsistency of individual model feature attributions versus the consistency of attributions ensembles of 15 for each tabular dataset in Figure 8 and Figure 9. The former shows inconsistency over differences in random initialization, the latter shows inconsistency over one-point changes to the training set.

## H.2 SIMILARITY METRICS OF ATTRIBUTIONS

We display how Spearman's ranking coefficient ($\rho$), Pearson's Correlation Coefficient ($r$), top-5 intersection and $\ell_2$ distance between feature attributions *over the same point* become more and more similar with increasing numbers of ensemble models. While the comparisons to generate the similarity score is between two models on the same point, the result is averaged over this comparison for the entire test set. We average this over 276 comparisons between different models. In cases were abstention is high, indicating inconsistency on the dataset for the training pipeline, selective ensembling can further improve stability of attributions by not considering unstable points (see e.g. German Credit). We present the expanded results from the main paper, for all datasets, on all four metrics (as SSIM is only computed for image datasets, and $\rho$ is not computed for image datasets). We display error bars indicating standard deviation over the 276 comparisons between two models for tabular datasets, and 40 comparisons for image datasets.

## I   INTUITION BEHIND ENSEMBLE GRADIENT CONSISTENCY

In section 4, we demonstrated how prediction inconsistency can be provably bounded in selective ensembles. Section 5.2 showed that ensembling also improves the consistency of the *gradients*; we now provide some theoretical insight as to why this is the case.

| | | | | | *accuracy (abstain as error) / abstention rate* | | | | | |
|---|---|---|---|---|---|---|---|---|---|---|
| $\mathcal{S}$ | $n$ | Adult Male | Adult Fem. | Ger. Cred. Young | Ger. Cred. Old | Tai. Cred. Male | Tai. Cred. Fem. | Warf. Black | Warf. White | Warf. Asian |
| Base | 1 | .804/ - | .923/ - | .677/ - | .777/ - | .814/ - | .825/ - | .665/ - | .688/ - | .689/ - |
| RS | 5 | 0.0/1.0 | 0.0/1.0 | 0.0/1.0 | 0.0/1.0 | 0.0/1.0 | 0.0/1.0 | 0.0/1.0 | 0.0/1.0 | 0.0/1.0 |
| RS | 10 | .777/.053 | .912/.023 | .507/.334 | .636/.254 | .791/.048 | .807/.035 | .659/.009 | .681/.002 | .683/.007 |
| RS | 15 | .786/.037 | .915/.015 | .559/.248 | .705/.168 | .798/.033 | .812/.025 | .664/.010 | .683/.002 | .688/.006 |
| RS | 20 | .789/.030 | .917/.013 | .586/.205 | .733/.130 | .802/.028 | .814/.020 | .667/.009 | .683/.002 | .689/.006 |
| Base | 1 | .806/ - | .922/ - | .697/ - | .757/ - | .815/ - | .825/ - | .665/ - | .687/ - | .688/ - |
| LOO | 5 | 0.0/1.0 | 0.0/1.0 | 0.0/1.0 | 0.0/1.0 | 0.0/1.0 | 0.0/1.0 | 0.0/1.0 | 0.0/1.0 | 0.0/1.0 |
| LOO | 10 | .787/.038 | .913/.018 | .612/.166 | .689/.138 | .802/.023 | .817/.014 | .655/.020 | .680/.019 | .680/.019 |
| LOO | 15 | .793/.026 | .916/.012 | .646/.101 | .704/.107 | .806/.017 | .819/.011 | .658/.014 | .681/.013 | .682/.013 |
| LOO | 20 | .796/.022 | .917/.010 | .661/.071 | .714/.084 | .808/.014 | .820/.009 | .659/.011 | .682/.011 | .683/.011 |

Table 9: We present the selective ensemble accuracy and abstention rate group-by-group for several different demographic groups across four datasets: Adult, German Credit, Taiwanese Credit, and Warfarin Dosing. We note that by and large, using selective ensembles did not exacerbate accuracy disparity by very much (within 1% of the original disparity), although they did not ameliorate disparities in accuracy that already existed within the performance of the algorithm. The only exception to this was German Credit, where we note, as in the remainder of our results, that the entire dataset is only 1000 points, so results may be slightly different in this regime. Overall, we note that subgroup abstention rates can vary by dataset, and so it should be studied whenever selective ensembles are used in a sensitive setting.

At a high level, we argue that by taking the average gradient (of the mode prediction, w.r.t. the input), we reduce the variance in the ensemble gradient, stabilizing it towards the expected gradient over the distribution $\mathcal{S}$. While it is difficult to exactly characterize the distribution over gradients, we provide a formalized intuition by making some simplifying assumptions about this distribution.

We model the variations in gradients from model to model as a Gaussian (i.e., the gradient of each model deviates from the expected gradient by some Gaussian noise). Formally, let $f = \mathcal{P}(S)$ be a model produced by the pipeline on some random state, $S$, and let $\hat{\nabla} f(x) = \mathbb{E}_{S \sim \mathcal{S}}[\nabla f(x)]$ be the expected gradient over $\mathcal{S}$. We will assume that $\nabla f(x)$, the gradient of a particular model produced by the pipeline, is given by $\hat{\nabla} f(x) + \eta$, where $\eta \sim \mathcal{N}(0, \sigma^2)$.

Under this assumption, the fact that the gradient is stabilized by ensembling follows from the fact that the variance of the sample mean is smaller than the population variance. Specifically, the variance of the sample mean of the gradient with $n$ samples is $\sigma^2/n$, which tends towards 0 as $n$ increases (Equation 6).

$$\text{Var}\left[\frac{1}{n}\sum_{i=1}^{n}\nabla f(x)\right] = \frac{\sigma^2}{n} \tag{6}$$

The metrics we measure for gradient consistency are not based on variance of the gradient (which Equation 6 shows is reduced by ensembling), but rather *ranking* of features according to their gradients. To relate this to variance we can argue the following: first, let us take, for example, the top-ranked feature (the feature with the largest positive gradient). Assuming the expected gradient is not degenerate (in our case this means there is a unique top-ranked feature), we can consider the gap between the average gradient of the top-ranked feature, $i$, and the second-ranked feature, $j$. When the variance in the ensemble gradient is reduced sufficiently relative to this gap, it will be unlikely that this order will be switched.

Specifically, let $\hat{\delta} = \hat{\nabla}_i f(x) - \hat{\nabla}_j f(x)$ be the gap between the average gradient of the top-ranked feature and the second-ranked feature, and let $\delta(n) = \frac{1}{n}\sum_{k=1}^{n}\nabla_i f(x) - \nabla_j f(x)$ be the gap between the gradient of the same features on an ensemble of $n$ models produced by the pipeline. Note that the ordering between features $i$ and $j$ is preserved provided that $\delta(n) > 0$. Thus, we can quantify the probability that this ordering is preserved according to Equation 7.

$$\mathbf{Pr}[\delta(n) > 0] = \mathbf{Pr}[\eta_i - \eta_j > 0] \qquad \text{where } \eta_i \sim \mathcal{N}\left(\hat{\nabla}_i f(x), \frac{\sigma^2}{n}\right) \text{ and } \eta_j \sim \mathcal{N}\left(\hat{\nabla}_j f(x), \frac{\sigma^2}{n}\right)$$

$$= \mathbf{Pr}\left[\hat{\delta} > \eta_{ij}\right] \qquad \qquad \text{where } \eta_{ij} \sim \mathcal{N}\left(0, \frac{2\sigma^2}{n}\right)$$

$$= \Phi\left(\hat{\delta}\sqrt{\frac{n}{2\sigma^2}}\right) \tag{7}$$

The same argument holds for other pairs of features.

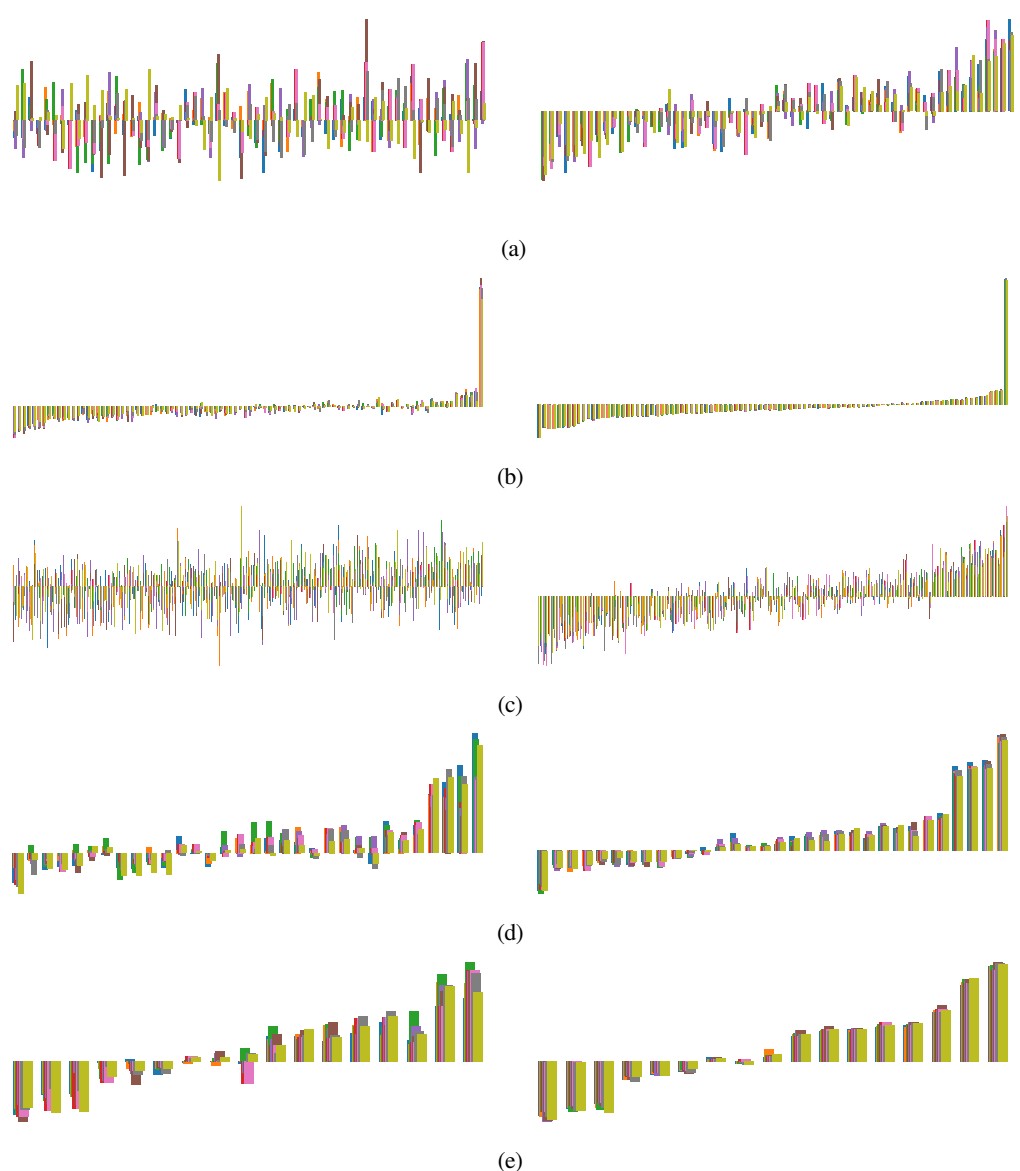

Figure 8: Inconsistency of attributions on the same point across an individual (left) and ensembled (right) model ($n = 15$), for all datasets, over differences in random seed chosen for initialization parameters before training. The height of each bar on the horizontal axis represents the attribution score of a distinct feature, and each color represents a different model. Features are ordered according to the attribution scores of one randomly-selected model. Figure a depicts the German Credit Dataset, Figure b depicts Adult, Figure c Seizure, Figure d Taiwanese, and Figure e Warfarin. We do not include feature attribution for image datasets as the individual pixels are less meaningful than the feature attributions in a tabular dataset.

## J   ACCURACY REJECTION CURVES

Figure 11 shows plots of accuracy-rejection curves Nadeem et al. (2009) for selective ensembles of sizes 5, 10, 15, and 20. Rejection rates were controlled by varying $\alpha$ from 1 to 0. In accordance with the convention of Nadeem et al., we count the accuracy as 1.0 when the model abstains on all points.

The plots indicate that accuracy remained relatively high even with low abstention rates and did not increase substantially when more points were rejected, signifying that there was not a strong trade-off between accuracy and rejection. This is desirable because we clearly prefer to have a low rejection rate; meanwhile,

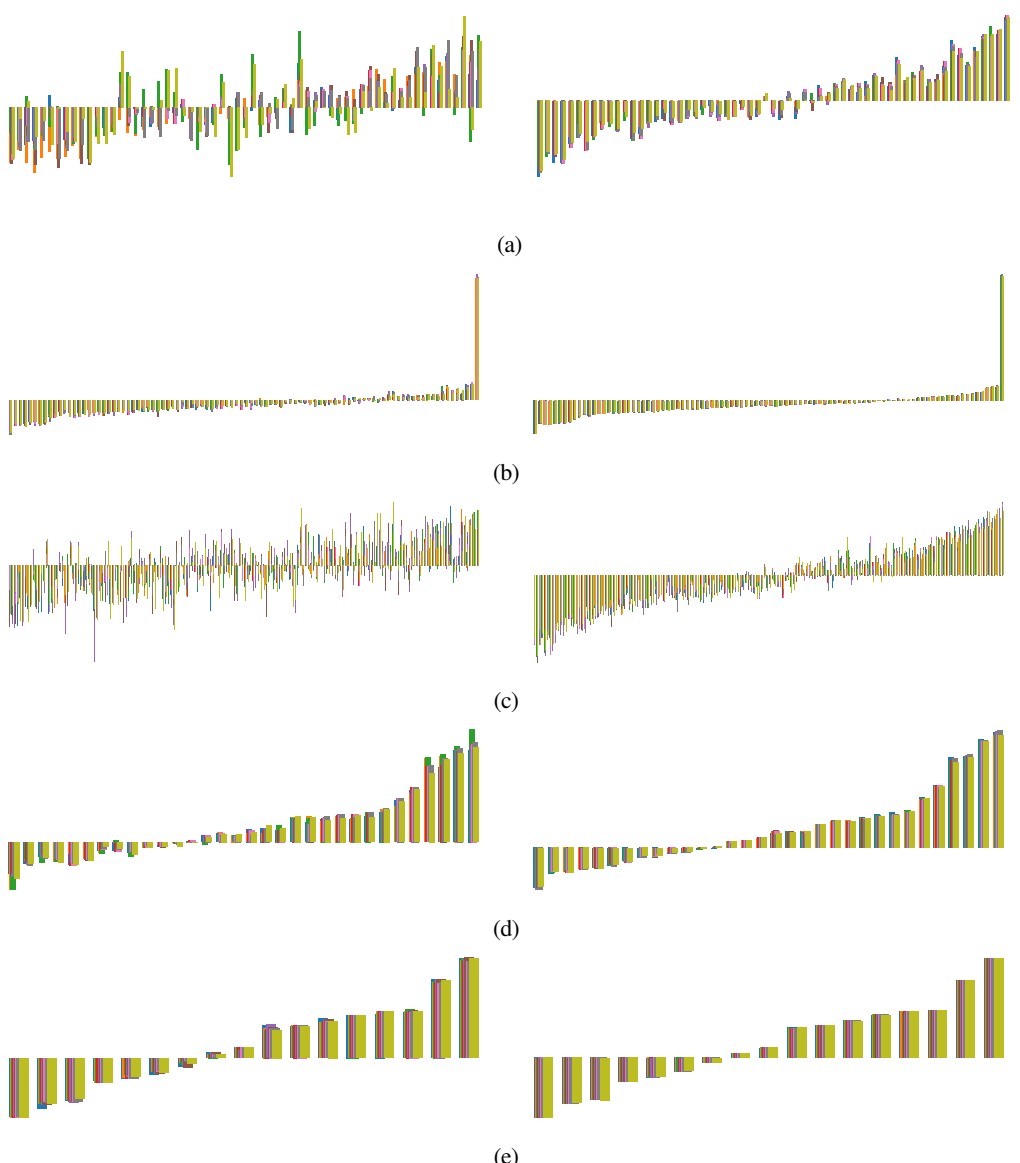

Figure 9: Inconsistency of attributions on the same point across an individual (left) and ensembled (right) model ($n=15$), for all datasets, over leave-one-out differences in the training set. The height of each bar on the horizontal axis represents the attribution score of a distinct feature, and each color represents a different model. Features are ordered according to the attribution scores of one randomly-selected model. Figure a depicts the German Credit Dataset, Figure b depicts Adult, Figure c Seizure, Figure d Taiwanese, and Figure e Warfarin. We do not include feature attribution for image datasets as the individual pixels are less meaningful than the feature attributions in a tabular dataset.

the purpose of abstention is to guarantee consistency, so we do not expect abstention to have a strong effect on accuracy. Furthermore, low rejection rates correspond to more consistent predictions. Finally, we note that rejection rates are kept low even for small values of $\alpha$ by increasing the size of the ensemble.

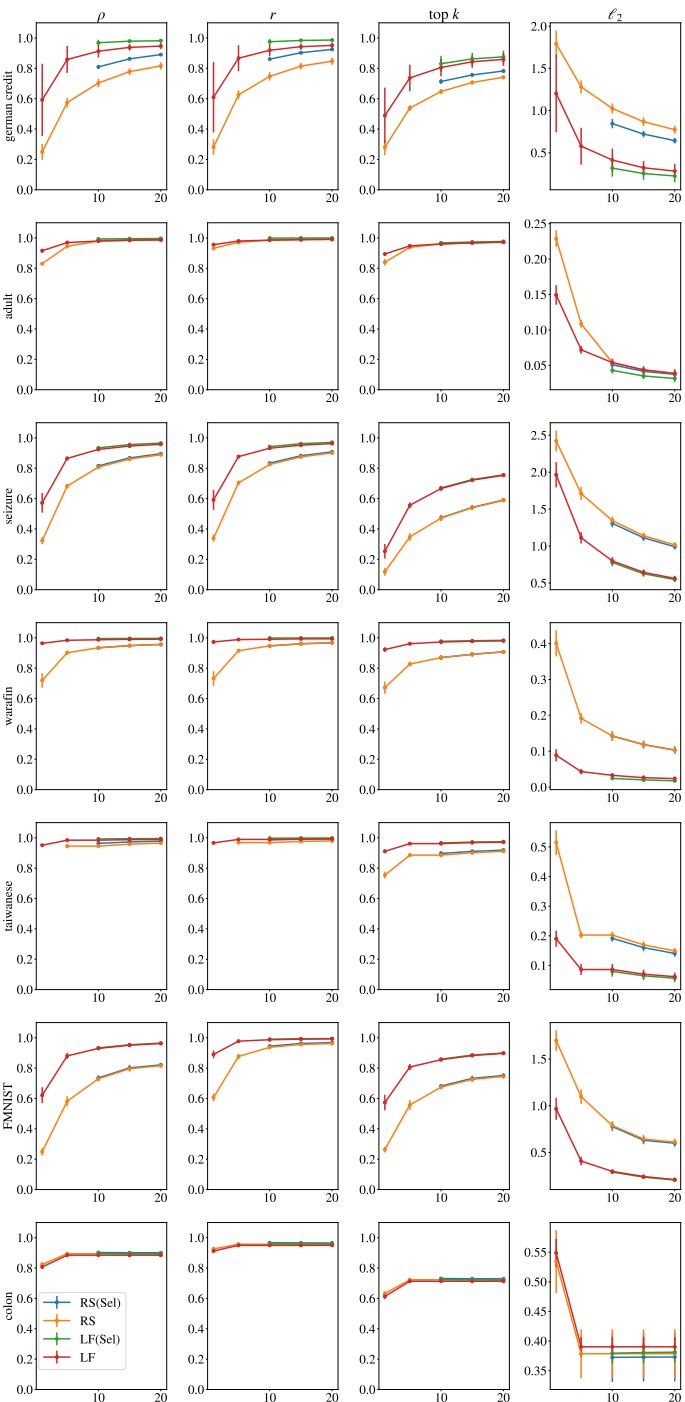

Figure 10: We plot the average similarity across feature attributions for an individual point, averaged over 276 comparisons of feature attributions from two different models. This is aggregated across the entire validation split. The error bars represent the standard deviation over the 276 comparisons between models. Each row of plots constitutes the plots for a given dataset, noted on the far left, and each column of plots is for a given metric, noted at the top. Note that for image datasets, (FMNIST and Colon), we plot SSIM instead of Spearman's Ranking Coefficient ($\rho$). The x-axis is the number of models in the ensemble, starting with one, and the y-axis indicates the value of the similarity metric averaged over all 276 comparisons of individual points' in the validation split's attributions. The red and orange lines depict regular ensembles, and the green and blue represent selective ensembles.

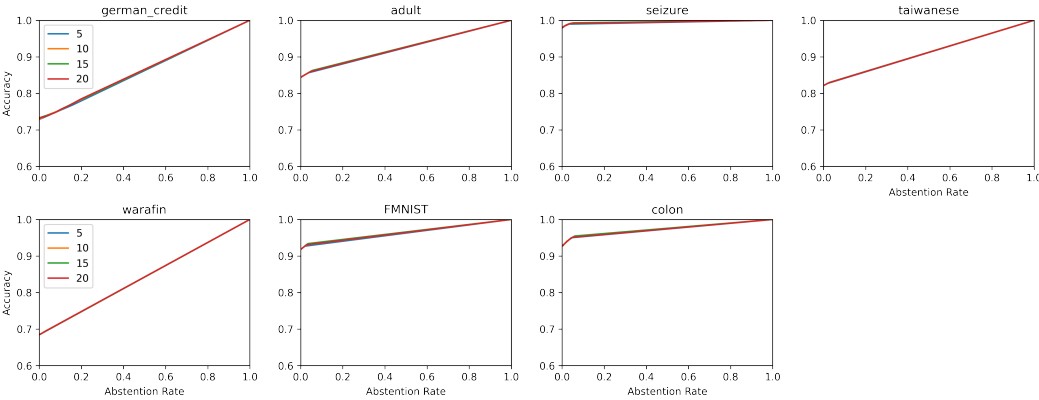

Figure 11: Graphs of accuracy (y) versus abstention (x) of ensembles of different size, gathered by calculating accuracy and abstention for 12 different values of $\alpha$. Note that the y-axis begins at 0.6.

