# OpenReview forum: "Selective Ensembles for Consistent Predictions"
_ICLR.cc/2022/Conference — ICLR 2022 Poster_

### Official Review · Reviewer_hSPx · 2021-11-02

**Correctness:** 4
**Technical Novelty And Significance:** 3
**Empirical Novelty And Significance:** 3
**Recommendation:** 8
**Confidence:** 4

**Main Review:**

The paper motivates a very practical problem of model and feature attribution instability, and proposed a solution with a plethora of theoretical and empirical guarantees. The theoretical development is sound, and the empirical study section is comprehensive and convincing. The paper is also well written in a clear and concise manner.

The reviewer's major concerns here are based on the novelty and the feasibility of the proposed approach. It is worth pointing out that neither of these two points prevents the paper from being considered as a highly completed work, and the reviewer would gladly update the rating should the concerns be concisely addressed.

1. Regarding the novelty, it is not very surprising to see that bagging / ensembling helps with the prediction stability for classifiers, especially when equipped with reject option so that the most ambiguous (hard) examples can somehow evade the strict metric computation. What seems more novel here is the improvement of gradient stability, which is still expected (since taking gradient is still a linear operation inside the expectation) but has involved less study. The empirical evidence shown in Figure 4 is strong, and it would be even better if there were corresponding theoretical discussions.
2. The proposed method constructs multiple (20 at most as in the empirical study) models. Since the discussion seems to be heavily focused on deep neural nets, two things to catch here:
- It is unclear whether the proposed method would feasibly apply to cases involving giant model structures and a large amount of output classes. (This comment is slightly nitpicking)
- On one hand, neural net ensembles can be technically constructed as a single neural net with an aggregating final layer and multiplicative numbers of parameters. On the other hand, neural nets themselves are showing behaviors of ensembles, especially when trained with dropout or similar mechanism. The proposed method does not take advantages of these existing relationships between neural nets and neural net ensembles.

The reviewer also finds two minor theoretical issues that might deserve a bit more attention
- The power of the top_2 test is highly influenced by the data generating distribution (e.g. a fair coin flip would also trigger the reject option, and maybe the German Credit dataset), therefore \alpha and \beta are probably not two independent parameters.
- For reject (abstention) options there is usually a penalty for saying "I'm not sure". It would influence the rates by a fraction of \beta which is safe to ignore.

The paper is nicely written. The only editorial comments:
1. Theorem 3.1 is also referred as Theorem A.1 in multiple places.
2. Table 3, LF -> LOO
3. Section 3, line 5, "Feature attributions are a commonly used as a tool...", redundant "a".



=====
The reviewer would like to upgrade the rating to an 8 after the author response.


**Summary Of The Paper:**

In the paper the authors proposed a selective ensemble whose constituents are classifiers with reject (abstention) option. They theoretically exposed the problem that differentiable models yielding identical predictions might remotely share identical gradients, biasing the feature attribution based on (integrated) gradient approaches. They theoretically demonstrated that the selective ensemble can contribute to model stability, and empirically showed that both the model stability and feature attribution could be improved as a result of the proposed method.

**Summary Of The Review:**

The paper presents a highly completed study on the improvements of model stability and feature attribution thanks to the contribution from selective ensembles with abstention option. The paper could use a bit more theoretical discussions regarding the inner workings of the proposed method as well as the relationship between the work and existing ensemble studies to highlight its novelty.

---

> ### Author Response · Authors · 2021-11-12
> **Thank you for your review**
>
> Thank you for your review. We are glad to hear that you found our work on gradient stability interesting. We will address your specific comments below.
>
> > What seems more novel here is the improvement of gradient stability, which is still expected (since taking gradient is still a linear operation inside the expectation) but has involved less study. The empirical evidence shown in Figure 4 is strong, and it would be even better if there were corresponding theoretical discussions.
>
> While it is difficult to exactly characterize the distribution over gradients, we could give a theoretical intuition by making some assumptions about this distribution. The main idea is that by taking the average gradient (of the mode prediction, w.r.t. the input), we reduce the variance in the ensemble gradient, stabilizing it towards the expected gradient over the distribution $\mathcal S$.
>
> If we model the variations in gradients as Gaussian (i.e., the gradient of each model deviates from the expected gradient by some Gaussian noise), this stabilization property follows from the fact that the variance of the sample mean is smaller than the population variance. For a distribution with variance $\sigma^2$, the variance of the sample mean with $n$ samples is $\sigma^2 / n$, which tends towards 0 as $n$ increases.
>
> The metrics we measure for gradient consistency are not based on variance, but rather ranking. To relate this to variance we can argue the following: first, let’s take for example the top-ranked feature (the feature with the largest positive gradient). Assuming the expected gradient is not degenerate (in our case this means there is a unique top-ranked feature), we can consider the gap between the gradient of the top-ranked feature and the second-ranked feature. When the variance in the ensemble gradient is reduced sufficiently relative to this gap, it will be unlikely that this order will be switched. The same argument holds for other pairs of features.
>
> We would be happy to include this discussion, either as a sketch like as we have described it here, or more formally in a set of lemmas. Please let us know what your preference would be, and we will update the paper accordingly.
>
> > Regarding the novelty, it is not very surprising to see that bagging / ensembling helps with the prediction stability for classifiers, especially when equipped with reject option so that the most ambiguous (hard) examples can somehow evade the strict metric computation.
>
> While the fact that ensembling aids prediction stability is expected (which we acknowledge in the paper given the long history of using ensembles for variance reduction), we are not aware of prior work which also provides an effective guarantee on consistency, and view this as a primary novelty of our work.

---

> > ### Author Response · Authors · 2021-11-12
> > **Response part 2**
> >
> > >It is unclear whether the proposed method would feasibly apply to cases involving giant model structures and a large amount of output classes. (This comment is slightly nitpicking)
> >
> > While the use of selective ensembles on large architectures would require more compute, we point out that the Colon dataset models in this paper are based on a ResNet50 (Appendix C discusses this, and we are happy to mention it in the main paper if it would be helpful). While we didn't test datasets with large numbers of classes, the amount of extra bookkeeping required to keep track of their frequencies is minimal compared to the size of the model and doesn't scale with added constituent models. Also note that the binomial test only needs to look at the top two classes, so the consistency guarantee need not degrade as the number of classes increases.
> >
> > > On one hand, neural net ensembles can be technically constructed as a single neural net with an aggregating final layer and multiplicative numbers of parameters. On the other hand, neural nets themselves are showing behaviors of ensembles, especially when trained with dropout or similar mechanism. The proposed method does not take advantages of these existing relationships between neural nets and neural net ensembles.
> >
> > We agree that finding methods to distill the increased stability from an ensemble into a single model is an interesting direction. However, we note all the analysis in our paper is centered on calculating the mode of a set of predictions across a randomness over a training pipeline P. The main challenge with making this work with the consistency guarantee is that the distillation process will constitute just one instantiation of the pipeline’s random state, and our test requires multiple samples. Thus, we believe distilling the consistency benefits from a selective ensemble into one model would be a fascinating, but non-trivial extension to pursue in future work.
> >
> > > The power of the top_2 test is highly influenced by the data generating distribution (e.g. a fair coin flip would also trigger the reject option, and maybe the German Credit dataset), therefore \alpha and \beta are probably not two independent parameters.
> >
> > Alpha and beta are not two independent parameters: the selection of $\alpha$ directly impacts $\beta$ (for a constant number of constituent models $h$) over a given distribution. We are happy to edit any part of the paper that suggests otherwise. However, we note that in non-degenerate cases (i.e. where the mode is well-defined), $\beta$ can be reduced while *increasing* $\alpha$, by increasing the number of constituent models in the ensemble.
> >
> > When the distribution is completely bimodal on an instance, or the mode is otherwise not well defined (e.g., as you say, the result of a fair coin flip), then our approach is likely to abstain. However, we argue that in this case, abstaining is the right thing to do---it should be interpreted as a signal that the multiplicity cannot be resolved with any confidence (indicating a deeper problem with the model class/pipeline/learning objective). In some cases, as prior work has argued [Ustun et al. 2019, Black and Fredrikson 2021] an uncertain (and therefore potentially inconsistent) decision can become a low-quality or even unjust decision. In such cases, a model may do better to refer such instances to another decision mechanism---such as a human.
> >
> > > For reject (abstention) options there is usually a penalty for saying "I'm not sure". It would influence the rates by a fraction of \beta which is safe to ignore.
> >
> > Would it be possible to clarify this a bit more? We are happy to address it, but need further clarification about how we could explain or otherwise incorporate this thought into the paper.

---

> > > ### Author Response · Authors · 2021-11-19
> > > **Update**
> > >
> > > Thank you again for your review. We have included a more formalized description of the intuition we provided about improving consistency of gradients via ensembling in Appendix I of the paper. We have also addressed your minor formatting comments.
> > >
> > > We are happy to continue our discussion if you have any follow up questions or additional comments or concerns.

---

### Official Review · Reviewer_dvdh · 2021-11-02

**Correctness:** 4
**Technical Novelty And Significance:** 3
**Empirical Novelty And Significance:** 2
**Recommendation:** 5
**Confidence:** 4

**Main Review:**

The topic is interesting and the algorithms nice and simple. However, the empirical section is quite difficult to read due to insufficient descriptive details in the captions, as well as out of order referencing in the main text.

strengths:

- Theoretically sound idea which is also simple in practice.

weaknesses:

- The authors demonstrate that selective ensembles remove disagreements by contrasting with standard ensembles, but don't show accuracy comparisons against standard ensembles. Table 1 could do with comparitive numbers from a standard ensemble.
- Captions in general don't explain the plots/tables sufficiently. Table 4 in particular is uninterpretable as it is unclear what the two numbers
in the cells actuall are. This is a significant issue for interpreting the feature attribution results.
- Resampling can also involve features not just samples, like the random forest (an ensemble method) has demonstrated. This is not explored in the paper.

minor:

- I'm used to captions above tables and below figure, so I kept reading the wrong thing with consecuitive tables
- Table 3/fig 3 seem to be discussed and referenced before table 1/fig 1; maybe swap the order
- figure 4 as a bar chart is visually misleading due to overplotting of many colours; why not just plot the points with some x displacement to prevent overlaps?

**Summary Of The Paper:**

The paper considers the problem of dealing with inconsistencies when forming (binary) classifier ensembles. The authors propose a mechanism whereby an ensemble can abstain from making a prediction based on a statistical test on the predictions from its constituents.

**Summary Of The Review:**

While an interesting method the lack of clarity in the empirical experiments reduces the impact. Furthermore, the method reduces disagreement by essentially avoiding the problem through abstention; it's not really demonstrated that this has a practical advantage other than through highlighting specific cases for manual intervention or inspection.

---

> ### Author Response · Authors · 2021-11-12
> **Thank you for your review**
>
> Thank you for your review. We apologize for any lack of clarity that made this submission difficult to read, and will remedy those problems. We clarify first some aspects of our method and results, and then note that we have updated several figures and tables in the paper with a longer description.
>
> > The authors demonstrate that selective ensembles remove disagreements by contrasting with standard ensembles, but don't show accuracy comparisons against standard ensembles. Table 1 could do with comparitive numbers from a standard ensemble.
>
> Thank you for pointing us to a typo in our experimental results section that probably led to some confusion. The reference to Table 1 in Section 5.2 is actually Table 2, and Table 1 is referenced in the paragraph above. (We have fixed this mistake). Note we do have a table similar to Table 2 (showing accuracies of non-selective ensembles) in the appendix. However we have updated Table 2 in the paper with the accuracies of non selective ensembles---if you find this preferable, please let us know, and we will persist this change.
>
> > Table 4 in particular is uninterpretable as it is unclear what the two numbers in the cells actually are. This is a significant issue for interpreting the feature attribution results.
>
> We have adjusted the descriptions of Table 4 and Figure 3 in the main paper with additional information. We are happy to provide additional context in the rest of the figures, but would appreciate pointers to which aspects of them were confusing. We have also changed the order of (what were) Tables 2 and 3, which are discussed in opposite order in the text.
> We are happy to follow up on any of these points through further discussion.
>
> >the method reduces disagreement by essentially avoiding the problem through abstention
>
> We would like to point out that abstention is not the solution to disagreement---the solution to model disagreement is predicting the mode prediction (as defined over a training pipeline $P$) over several models. As we see, predicting the mode through ensembling leads to lower inconsistency (6% instead of 26% on German Credit , 2% instead of 8% on Adult), even with no abstention. Abstention is the method by which we lead to a theoretical *guarantee* of consistency, it ensures that the predictions are in fact the mode and thus will be consistent. Our experiments show that the abstention rate declines as more models are included, which suggests that it is not abstention that leads to consistency, but rather the ability to predict the mode.
>
>
> >it's not really demonstrated that this has a practical advantage other than through highlighting specific cases for manual intervention or inspection
>
> Our work rigorously addresses the problem of inconsistency in model predictions and explanations over retrainings, which has been identified in multiple instances of prior work  [D’Amour et al. 2020, Black and Fredrikson 2021, Pawelczyk et al. 2020, Rawal et al. 2020, Marx et al. 2019]. Consistency in predictions and explanations is of importance in several applications, from criminal justice to personal finance to medicine, due to model usability, procedural justice, and fairness concerns, as has been argued in previous work [Black and Fredrikson 2021, Marx et al. 2019, Rawal et al. 2020]. Selective ensembles mitigate this problem by guaranteeing consistent predictions with low abstention rates (generally between 1-3%). In cases where even this small amount of abstention is unacceptable, but high-quality, consistent answers are desired, as you say, this method can highlight specific cases for manual intervention. We argue this is in fact an important quality, as it allows for individuals who may have been given an inconsistent, and therefore perhaps low-quality or otherwise unfair outcome to be given a higher quality outcome through another decision process.
>
> Thank you very much for your review, and we look forward to continued discussion should you have follow up questions or comments.

---

### Official Review · Reviewer_AzPU · 2021-11-02

**Correctness:** 2
**Technical Novelty And Significance:** 3
**Empirical Novelty And Significance:** 2
**Recommendation:** 5
**Confidence:** 2

**Main Review:**

Although the idea is interesting and theoretically sound, it is unclear in which aspect the proposed method is superior to existing methods.
- If it's classification accuracy, the method should be compared with such methods as hard-voting prediction, soft-voting prediction, and other basic ensemble methods that aggregate individual predictions, and also individual model's prediction and MC-dropout.
- If it's uncertainty quantification performance (or selective classification performance - trade-off between classification accuracy and abstention rate), various uncertainty quantificaiton measures can be compared, such as entropy, BALD, .... Deep ensemble can also be compared.
- Lakshminarayanan, B., Pritzel, A., & Blundell, C. (2016). Simple and scalable predictive uncertainty estimation using deep ensembles. arXiv preprint arXiv:1612.01474.

Another contribution the authors mentioned is that the proposed method achives consistent feature attribution. i'm wondering what's the practical meaning of it. how does it relate to the use of a model in classifying data?

For the proposed method, the absention rate may be adjustable by using different criteria for the statistical testing. It would be interesting if the proposed method is evaluated in terms of accuracy-rejection curve to see the trade-off between classificaiton accuracy and abstention rate)
- Nadeem, M. S. A., Zucker, J. D., & Hanczar, B. (2009, March). Accuracy-rejection curves (ARCs) for comparing classification methods with a reject option. In Machine Learning in Systems Biology (pp. 65-81). PMLR.

Minor comments
please elaborate on what "binom_p_value", "top_2" function, n_A, and n_B are in Algorithm 2 to help readers to comprehend.


**Summary Of The Paper:**

This paper presents a selective ensemble method. Given a set of models, it performs a statistical test to determine if the mode prediction was selected by a statistically significant majority of the constituent models. If the test succeeds, it returns the mode prediction, Otherwise, it abstains from prediction. the authors demonstrated the effectiveness of their method on improving classification accuracy by abstaining from those that are difficult to classify.

**Summary Of The Review:**

It's unclear how the proposed method is beneficial in classification problems. The authors should provide the practical meaning of their method and compare their method with some proper baselines if necessary.

---

> ### Author Response · Authors · 2021-11-12
> **Thank you for your review**
>
> Thank you for your review. We first clarify the goals of our work, and then specifically describe how they are different from both increasing classification accuracy or creating uncertainty predictions.
>
> ### Goals
> Our goal in this paper is not to increase classification accuracy or create accurate uncertainty estimates, but to build a model that returns consistent (even if sometimes incorrect) predictions over changes to its training environment (e.g., training with a different random seed). That is, the purpose of a selective ensemble is to return the same prediction across random retrainings from the training pipeline $P$. We achieve consistent predictions by predicting the mode outcome across the set of models that could result from a training pipeline $P$ with a given source of random variation. To provide a theoretical guarantee on the model’s consistency, we allow the ensemble to abstain when it is not certain that its prediction is the true mode.
>
> ### Why consistency?
> As models deployed for use in real-world contexts are usually retrained over time (updates to hyperparameters, or training set, etc.), they may change their predictions or explanations due to these re-trainings---as prior work [D’Amour et al. 2020, Black and Fredrikson 2021], as well as our own results, demonstrate. Meanwhile, consistency is important in many contexts due to model usability, procedural justice, and fairness concerns, as has been argued in previous work [Black and Fredrikson 2021, Marx et al. 2019, Rawal et al. 2020]. For example, inconsistent predictions may jeopardize applicants’ possibility of recourse (i.e. changing their application to be accepted at a later date based on insights or explanations from the model at their time of initial application). We focus on consistency in this paper to address these problems.
>
> ### Difference from accuracy maximization
> We do not target accuracy maximization through selective ensembles, but rather consistency. Consistency guarantees are obtained by targeting the mode predictor, which is fixed by the training pipeline, $P$. If the training pipeline is well-suited to the task, the mode will be accurate---if the mode tends to be *inaccurate*, then this may call for changes to the pipeline (e.g., changing the model class or training procedure for the constituent models), however this is not the problem we address. We focus on making the ensemble’s predictions consistent in this paper, taking the pipeline $P$ as given, and assuming the mode predictor has sufficient accuracy.
>
> Thus, relevant comparisons to selective ensembles are modeling methods which reduce inconsistency (rather than accuracy) over a *fixed* pipeline $P$. As we note in the paper, ensembling in general reduces inconsistency as it has long been used as a method of variance reduction. However, our method is unique in that it provides a theoretical *guarantee* on consistency---to our knowledge, this is the first paper to provide such a guarantee, so we do not see any clear comparisons for this metric (consistency guarantees). Making changes to the training pipeline $P$ to increase accuracy (or *empirical* consistency) is complementary, but orthogonal, to this work.
>
> ### Difference from Uncertainty Quantification
> We interpret “uncertainty estimation” in this context to mean obtaining a probability that a model’s prediction is incorrect, as in the referenced deep ensemble paper (please let us know if we misunderstood what you meant). Uncertainty estimation is thus relative to the *correct prediction*, whereas the consistency sought by selective ensembles is relative to the *mode prediction* across models generated by a given pipeline. The guarantee provided by our approach bounds the probability of predicting differently than the mode, and does not attempt to quantify the probability that this is the correct prediction. Thus, these are different considerations.
>
> Given these differences, selective ensembles cannot be directly compared to uncertainty estimation. However, if the uncertainty estimation can be *guaranteed to be calibrated*, it could be used as a prediction threshold to bound inconsistent predictions, if we allow for abstention. If we abstain whenever the model is not highly confident (e.g., 95% confidence), then we would be able to bound the probability of disagreement whenever two models both do not abstain on a point, giving us a similar guarantee to selective ensembles. However a few problems exist even using this approach with, e.g., deep ensembles. (1) Deep ensembles do not give a guarantee that their uncertainty estimates will be calibrated, so we will not truly get a guarantee. (2) If we take the uncertainty estimate at face value, (e.g., at 95% confidence, which would give a slightly worse guarantee than $\alpha = 0.05$ in our context) the rejection rate will likely be quite high (we ran some preliminary experiments and found a rejection rate of 30-50%).

---

> > ### Author Response · Authors · 2021-11-12
> > **Response part 2**
> >
> > ### Other methods
> > Some other modeling methods might empirically reduce inconsistency. In general, our approach is complimentary to other modeling methods which increase consistency: models which are inherently more consistent can be used as constituent models in a selective ensemble, which can be used to provide a guarantee on consistency. That is, creating models which are empirically more consistent can be seen as making changes to the training pipeline $P$ of a selective ensemble, which is complimentary but orthogonal to this work.
> >
> >
> > > Clarification on “binom_p_value", "top_2" function, “n_A”, and “n_B”
> >
> > In order to answer these questions, we review the selective ensemble prediction algorithm. To predict on a point $x$, a selective ensemble:
> > * obtains the classification (i.e., not the underlying probability) prediction from all constituent models $h$ in the ensemble.
> > * Then counts how many “votes” there are for each class among the models. The top_2 function finds which classes have the top 2 number of votes. We call these classes $A$ and $B$, with number of votes $n_A$ and $n_B$.
> > * It then uses the binomial test---a two-sided test of the null hypothesis that the probability of success in a Bernoulli experiment is 50%---to see if the votes for class $A$ constitute a strong majority, suggesting it is indeed the mode prediction over the set of models created with pipeline $P$. (This test checks if the underlying probability of success of $n_A$ successes from $n_A+n_B$ trials is greater than 50%).
> > * If the p value for this experiment is small, (less than alpha), then this means that the prediction is likely to be the mode with probability 1-$\alpha$, and the model returns class $A$ as its prediction. Otherwise, it abstains.
> >
> > It is the use of this statistical test that allows us to bound the prediction inconsistency between any two selective ensemble models trained with pipeline $P$. As we show, selective ensembles can reach 0% prediction inconsistency across models, while keeping abstention to approximately ~1.5%.
> > We can add a fuller explanation of this algorithm in the description of the algorithm, and produce a paragraph similar to this in the appendix.
> >
> > > Another contribution the authors mentioned is that the proposed method achieves consistent feature attribution. i'm wondering what's the practical meaning of it. how does it relate to the use of a model in classifying data?
> >
> > Feature attributions are often used as explanation techniques, e.g. Integrated Gradients, among others (Simonyan et al., 2014; Sundararajan et al., 2017). If a classification model were to be used in a context where it was required to produce outputs for its explanations, for example, in loan application decision scenarios, where explanations are required by law in the US [ECOA], and EU [GDPR 2018], inconsistency in feature attributions would translate to inconsistency in explanations. This inconsistency could lead to confusion or distrust for those working with models, or subject to model decisions (and thus receiving model explanations). While instability in model explanations will not always degrade accuracy, it may degrade model usability through these other factors. We are happy to discuss this further if this does not satisfy the answer to your question.
> >
> > > For the proposed method, the abstention rate may be adjustable by using different criteria for the statistical testing. It would be interesting if the proposed method is evaluated in terms of accuracy-rejection curve to see the trade-off between classification accuracy and abstention rate)
> >
> > Yes, the abstention rate (up to a point) will be influenced by the choice of $\alpha$, i.e. the p-value required from the statistical test, and the underlying agreement between the models on the distribution. Note that we provide a figure describing the relationship between $\alpha$, the abstention rate $\beta$, and the agreement between models in Figure 2. If you are interested in an empirical exploration of the relationship between accuracy and abstention, are happy to provide approximated accuracy-abstention rate curves for 5 different selections of $\alpha$ on each dataset. We will update this response when those results are completed.
> >
> > Thank you again for your review, and we are happy to continue this discussion should there be follow-up questions or clarifications.
> >
> > References:
> >
> > 2018 reform of EU data protection rules. European Commission, Nov 11 2021. URL: https://ec.europa.eu/commission/sites/beta-political/files/data-protection-factsheet-changes_en.pdf.
> >
> > Equal Credit Opportunity Act. Bureau of Consumer Financial Protection. Nov 11 2021. URL: https://www.ftc.gov/enforcement/statutes/equal-credit-opportunity-act

---

> > > ### Author Response · Authors · 2021-11-19
> > > **Update**
> > >
> > > Thank you again for your review. We are updating to show our results from creating accuracy/rejection tradeoff curves, which are included in Appendix J of the updated paper.
> > >
> > > The plots indicate that accuracy remained relatively high even with low abstention rates and did not increase substantially when more points were rejected, signifying that there was not a strong trade-off between accuracy and rejection. This is desirable because we clearly prefer to have a low rejection rate.
> > >
> > > Note that in the context of uncertainty estimation, abstention directly allows the predictor to only select the points that are most likely to be correct---meanwhile, in our case, the purpose of abstention is to guarantee *consistency*, so we do not expect to have the same direct relationship between abstention and accuracy. Finally, any notion of a tradeoff between accuracy and abstention can be sidestepped by adding more models to the selective ensemble---so we do not need to trade lower accuracy for a lower abstention rate, but may simply add more models to the ensemble to improve both of these metrics.
> > >
> > > We are happy to discuss any follow up questions you may have.

---

### Official Review · Reviewer_5ZWm · 2021-11-04

**Correctness:** 4
**Technical Novelty And Significance:** 3
**Empirical Novelty And Significance:** 3
**Recommendation:** 6
**Confidence:** 2

**Main Review:**

The authors address the problem, where models trained for the same purpose achieve similar accuracy on consistent test data but behave very differently in individual predictions, is highly problematic and essential in domains such as medicine and finance where a high degree of reliability and stability is required.
To address this problem, the authors introduce a selective ensemble that mitigates such inconsistencies by applying hypothesis testing to the predictions of a set of models trained with randomly selected starting conditions.
In the selective ensemble, predictions are aborted when the constituent models disagree sufficiently. To demonstrate the effectiveness of the proposed selective ensemble, we first prove theoretically that the prediction discrepancies between the selective ensembles are bounded. Furthermore, using seven benchmark datasets, we show that the selective ensemble has zero points with inconsistent predictions and a low abstention rate of 1.5%.
The proposed idea is simple but appears to work well for the problem being tackled.

**Summary Of The Paper:**

In this paper, the authors address the problem that models trained for the same purpose may achieve similar accuracy on consistent test data but differently on individual predictions. Such inconsistent behavior is highly problematic, especially in the medical and financial domains. To address this problem, the authors introduce a selective ensemble that mitigates such inconsistencies by applying hypothesis testing to the predictions of a set of models trained on randomly selected starting conditions. First, they proved that the inconsistency of predictions between the selective ensembles is bounded. Additionally,  they empirically showed that the selective ensembles achieve consistent predictions and feature attributions while maintaining a low abstention rate. On seven benchmark datasets, the selective ensembles have zero points with inconsistent predictions. And an abstention rate as low as 1.5%. Thus, the selective ensemble proposed by the authors may present a more reliable method for using deep models in environments where high model complexity and stability are required.

**Summary Of The Review:**

This paper addresses an important problem in machine learning that is of great interest to many researchers and practitioners. The authors propose a method called selective ensembles and evaluate its effectiveness from both theoretical and empirical perspectives to solve this problem. Experiments on benchmark data show that the proposed method seems to work well for the problem. In conclusion, we believe that this paper is worthy of acceptance.

---

> ### Author Response · Authors · 2021-11-12
> **Thank you for your review**
>
> Thank you for your review. We are happy to answer any questions you may have should that be helpful.

---

### Author Response · Authors · 2021-11-19
**Update**

Thank you all for your reviews.

We have updated the papers with the suggestions by the reviewers:
* We have clarified descriptions of Tables and Figures throughout the evaluation sections
* We have added a theoretical analysis to motivate why ensembles improve gradient stability in Appendix I
* We have added tradeoff curves between accuracy and abstention of selective ensembles, following Nadeem et al. 2009, in Section J of the appendix.

---

### Decision · Program_Chairs · 2022-01-20

**Decision:**

Accept (Poster)

**Comment:**

This is a well-done job which combines a few ideas to reach means to identify problematic cases and indicate this when classifying. It has raised doubts about the applicability, though I can see that a abstention rule can have multiple uses. While the work seems to be well done, it has not largely excited the committee members. It initially missed to be placed well wrt existing work to highlight the novelty, and the demonstration that the approach can be generally useful is not complete. Dealing with abstention rules always brings another facet to classification and comparisons are not trivial in many situations. Not surprisingly, this has landed as a borderline case, which I place on the inside (as I like the topic and I think it is interesting work) but it could become an outsider depending on the overall view of the selected papers for the conference and other constraints.